# VIDEO2DEMO: GROUNDING VIDEOS IN STATE-ACTION DEMONSTRATIONS

## ABSTRACT

Vision-language demonstrations provide a natural way for users to teach robots everyday tasks. However, for effective imitation learning, these demonstrations must be perceptually grounded in the robot's states and actions. While prior works train task-specific models to predict state-actions from images, these often require extensive manual annotation and fail to generalize to complex scenes. In this work, we leverage pre-trained instruction-following Vision-Language Models (VLMs) that have shown impressive zero-shot generalization for detailed caption generation. However, VLM captions, while descriptive, fail to maintain the structure and temporal consistency required to track object states over time. We propose a novel approach, VIDEO2DEMO, that uses GPT-4 to interactively query a generative VLM to construct temporally coherent state-action sequences. These sequences are in turn fed into a language model to generate robot task code that faithfully imitates the demonstration. We evaluate on a large-scale human activity dataset, EPIC-Kitchens, and show that Video2Demo outperforms pure VLM-based approaches, resulting in accurate robot task code.

## 1 INTRODUCTION

Vision-language demonstrations offer a natural and powerful way for users to program robots to perform everyday tasks. However, for robots to imitate these demonstrations, they must be grounded in the robot's states and actions. Such grounding enables robots to abstract away the intricate details of their surroundings, focusing instead on higher-level logical reasoning (Konidaris et al., 2018; Mao et al., 2019; Xu et al., 2019; Garrett et al., 2021). Consider the example in Fig. 1, where a user demonstrates making and serving curry. Our goal is to extract high-level symbolic states, e.g. `is_touching('spoon', 'curry')`, and actions, e.g. `serve('curry')`, that can abstractly represent the task. This can in turn be fed into a language model that generates robot task code (Liang et al., 2022; Wang et al., 2023), enabling the robot to learn from the demonstration.

Prior work that grounds images in robot state-actions all rely on significant human annotation effort. One class of approach requires directly labeling states, which is intractable given combinatorially large state spaces (Mao et al., 2019; Lu et al., 2016; Newell & Deng, 2017; Yang et al., 2017). Others use weak supervision, like annotating actions (Ahmadzadeh et al., 2015; Migimatsu & Bohg, 2022). However, these rely on manually defining PDDL specification, which fails for large, complex, open-vocabulary datasets like EPIC-Kitchens (Damen et al., 2018; 2022).

Recent advances in Vision-Language Models (VLMs), such as PaLM-E (Driess et al., 2023), CLIP (Radford et al., 2021), Flamingo (Alayrac et al., 2022), train the model on large corpuses of multi-modal data and show impressive zero and few-shot generalization on tasks like caption generation and visual question answering. However, VLM-generated captions, while descriptive, fail to extract salient states from images and enforce temporal consistency over time.

***Our key insight is to frame the grounding problem as that of interactive visual question answering.*** We propose a novel approach, VIDEO2DEMO, that utilizes GPT-4 to query a VLM in an interactive manner. At every timestep, GPT-4, with its strong reasoning capabilities, asks the VLM questions about salient objects in an image and changes in their states. The VLM, in-turn, responds with detailed descriptions that update GPT-4's internal estimates and cause it to ask new questions. Through this dynamic back-and-forth of questions and responses, GPT-4 continually gathers salient

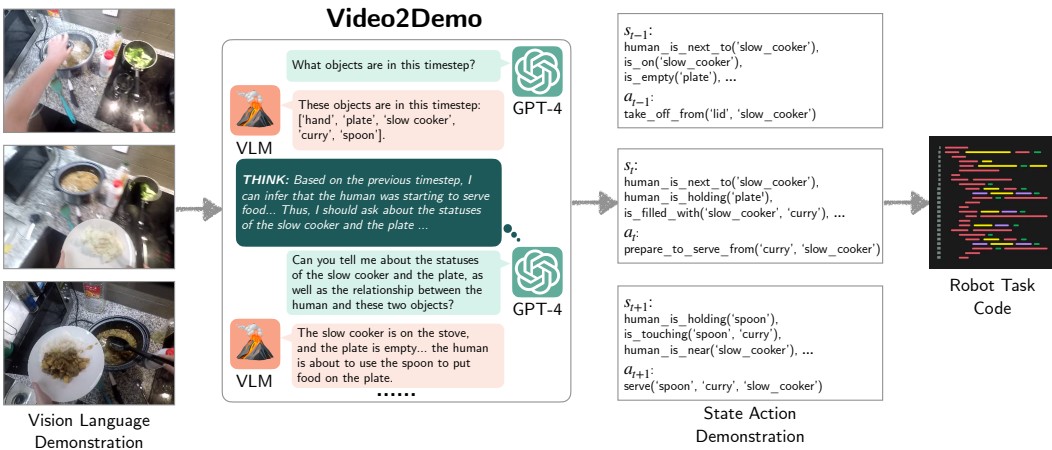

Figure 1: VIDEO2DEMO converts vision-language demonstrations to state-action sequences. In this framework, for each timestep, GPT-4 first interactively queries a VLM to extract salient information from each image. Then, it uses this information, along with predictions from previous timesteps, to predict the current states and action.

information over a number of rounds. Finally, GPT-4 uses the extracted information, along with its predictions from previous timesteps, to predict the current states and action.

Our key contributions are:

1. A novel LLM-VLM framework where GPT-4 interactively queries a VLM to transform vision-language demonstrations into temporally consistent, open-vocabulary state-action sequences.

2. Human-annotated state and action predicates for validation data in the large-scale human activity dataset, EPIC-Kitchens (Damen et al., 2018; 2022), which serves as a benchmark for state-action generation.

3. Evaluation using EPIC-Kitchens on both state-action generation and code generation with analysis showing that VIDEO2DEMO outperforms alternative LLM-VLM baselines.

## 2 RELATED WORK

### 2.1 VISION LANGUAGE GROUNDING

Symbolic state detection and action recognition are well established problems in computer vision. Symbolic state detection typically falls under the banner of visual relationship detection and scene graph generation (Lu et al., 2016; Newell & Deng, 2017; Xu et al., 2017; Yang et al., 2017; Liang et al., 2017; Zhang et al., 2017; Yang et al., 2018; Kolesnikov et al., 2019; Mao et al., 2019; Inayoshi et al., 2020). These techniques have recently been extended to action recognition, where symbolic states are manipulated over time (Ji et al., 2020). More broadly, these problems can be subsumed under the rubric of visual question answering (VQA) (Johnson et al., 2016; Goyal et al., 2017; Li et al., 2021; Agrawal et al., 2016) that focus on grounding visual understanding of models by asking questions about which objects are present in the scene, relation of objects to each other, colors/statuses of objects, etc. We cast our problem as an instance of VQA and leverage baseline vision-langauge models that have shown impressive performance.

There's been a surge of recent work on training joint text and image/video representations (Donahue et al., 2015; Fang et al., 2015; Vinyals et al., 2015; Wang et al., 2018; Yu et al., 2016; Miech et al., 2019; Zhang et al., 2023a; Gan et al., 2022; Li et al., 2023b) with models like CLIP (Radford et al., 2021) currently setting state-of-the-art performance. Multi-modal large language models (Driess et al., 2023; Huang et al., 2023a; OpenAI, 2023) trained on language and image data have shown impressive performance on visual question answering, captioning, and visual manipulation planning. Many recent work leverage open-source large-language models (Chiang et al., 2023; Touvron et al.,

2023a;b; Taori et al., 2023) to better align image and text embeddings (Liu et al., 2023; Li et al., 2023a; Zhu et al., 2023; Dai et al., 2023; Gao et al., 2023). Instruction tuning (Zhang et al., 2023b) over image-text parallel data improves the descriptive capabilities of these VLMs. In this work, we use LLaVA (Liu et al., 2023), a generative VLM fine-tuned on large amounts of image-conversation paired data generated by GPT-4 (OpenAI, 2023) from the COCO (Lin et al., 2015) dataset.

## 2.2 CONTROLLING ROBOTS VIA MULTIMODAL INPUT

Prior work in imitation learning from few-shot multi-modal demonstrations primarily focus on learning low-level sensory motor policies. These techniques adapt to the new scenario by conditioning on a robot demonstration (Finn et al., 2017; James et al., 2018), a video of a human (Yu et al., 2018; Bonardi et al., 2020), a language instruction (Stepputtis et al., 2020; Lynch & Sermanet, 2020), or a goal image (Pathak et al., 2018). Recent techniques look at learning skills by conditioning both on language and video demonstrations (Jang et al., 2022). However, these techniques train simple, visuomotor policies that are limited to short-horizon skills. Instead, we focus on learning complex, long-horizon tasks by leveraging the reasoning power of Large Language Models (LLMs).

Recent work have shown promising success in leveraging LLMs to control robots from language (Brohan et al., 2023; Ahn et al., 2022; Driess et al., 2023; Huang et al., 2022; Jiang et al., 2023; Singh et al., 2022; Huang et al., 2023b; Yu et al., 2023; Lin et al., 2023; Wu et al., 2023). However, many tasks require the use of both vision and language. This requires grounding images in the robot states. Traditional techniques (Ahmadzadeh et al., 2015; Migimatsu & Bohg, 2022) for grounding the visual information require humans to specify the set of possible predicates ahead of time as a PDDL specification, which is both tedious and difficult to scale in open-vocabulary settings. Instead, we leverage VLM's zero-shot capabilities to answer detailed questions about the scene and LLM's strong reasoning capabilities to infer predicates from VLM's responses.

The closest works in structure to our method are Socratic Models (SM) (Zeng et al., 2022), which leverage LLMs to invoke VLMs and other models, and NLMap (Chen et al., 2023), which is an instance of a SM that use VLMs to create a scene graph that LLMs can query. While SMs specify the interactions between models as a fixed computational graph engineered at the prompt level per task, our approach allows GPT-4 to direct the interaction, making it dynamic and contextual.

## 3 PROBLEM FORMULATION

We formalize each task as a Markov Decision Process (MDP). The state $s \in S$ is the set of all objects in the scene and their propositions, e.g. `is_open(obj)`, `is_inside(obj1, obj2)`, etc. We note that $S$ is combinatorially large, hence we avoid enumerating it explicitly. The action $a \in A(s)$ is a high-level operation of objects, e.g. `pick_up(obj)`, `place_down_at(obj, loc)`, where $A(s)$ is the affordance, i.e., the set of available actions at a state. The transition function $\mathcal{T}(\cdot|s, a)$ specifies how object states change under actions. The reward $R(s, a)$ captures a set of subgoals that define the task. The final goal is to learn the optimal policy $\pi : S \rightarrow A$ that maximizes total reward.

In the imitation learning setting, the user does not specify the rewards but instead provides demonstrations that show how to perform the task optimally. We assume that a demonstration is a sequence of egocentric images $\{o_1, o_2, \ldots o_T\}$ of the user doing the task, as well as an optional language narration $l$ describing the task. Both image and language play complimentary roles. The image $o_t$ is a partial observation on the true state of the environment $s_t$. The language narration $l$ describes the overall structure of the policy $\pi$ that solves the task.

We decouple the overall problem into two subproblems. The first subproblem, which we address in this paper, is to infer the most likely sequence of state-actions given image observations:

$$\underset{s_1, a_1, \ldots, s_T, a_T}{\arg\max} \quad \log P(s_1, a_1, \ldots, s_T, a_T | o_1, \ldots, o_T) \tag{1}$$

The second subproblem is that of imitation learning, i.e., mapping the state-acton sequence to a policy $\pi$ that imitates it. In our application, we represent policy $\pi$ as robot task code and leverage prior work (Wang et al., 2023; Liang et al., 2022) that predicts code from demonstrations represented in language form.

Problem 1 presents a number of challenges:

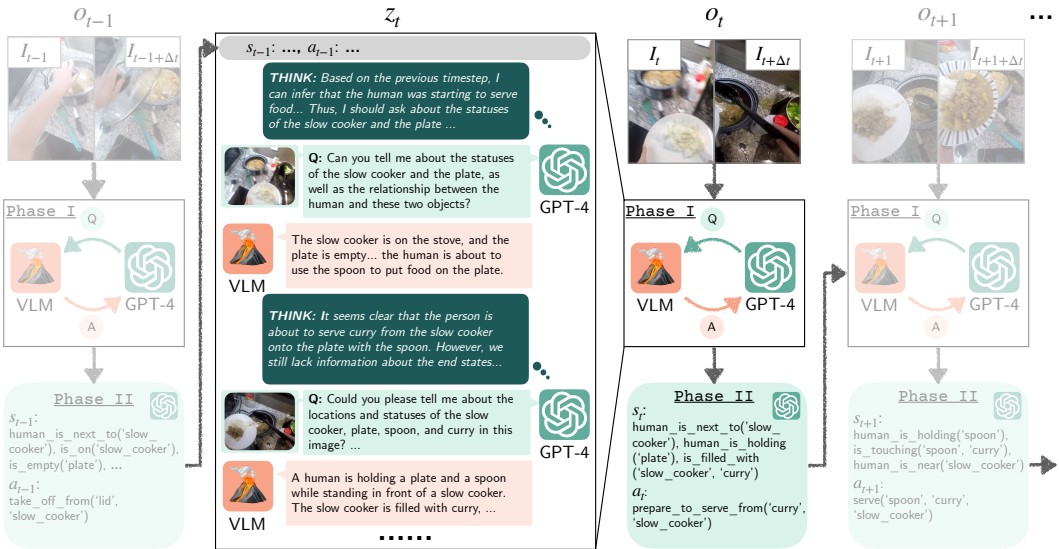

Figure 2: Overview of VIDEO2DEMO. At each timestep $t$, GPT-4 interactively queries the VLM about observation $o_t$ to extract information $z_t$. It then uses $z_t$, along with previous predictions $(s_{t-1}, a_{t-1})$, to predict the current state $s_t$ and action $a_t$.

1. **Complex Scenes with Irrelevant Objects:** Each image in $o_t$ represents scenes from real-world kitchen environments that can have many objects, but the state $s_t$ that is relevant for the task is a much smaller subset. Such a low signal-to-noise ratio poses challenges for the model.

2. **Partial Observability:** States $s_t$ may not be observable due to limited field of view, occlusion, and motion blur. Accurate estimation requires reasoning over multiple timesteps at a time.

3. **Generalization across Open-Vocabulary Tasks:** Finally, we would like the model to generalize across different tasks, environments, users, image conditions, etc. It should operate in open-vocabulary settings where it must identify new objects and states that it has not seen before.

Next, we tackle these challenges by combining the ability of Vision Language Models to generalize and handle complex scenes with the reasoning power of LLMs to solve this inference problem.

## 4 APPROACH

We present our framework, VIDEO2DEMO, that grounds vision-language demonstration in state-action sequences. Our key insight is to frame this grounding problem as *interactive visual question answering*, where GPT-4 interactively queries a VLM model, LLaVA (Liu et al., 2023), to infer states and actions. Fig. 2 shows an overview of VIDEO2DEMO, which contains two phases that are repeated for every timestep. In Phase I, GPT-4 holds an interactive conversation with LLaVA to extract salient information from the image. In Phase II, GPT-4 uses this extracted information, along with past predictions, to predict the current state and action.

### 4.1 PHASE I: INFORMATION EXTRACTION VIA INTERACTIVE VQA

In this phase, GPT-4 engages in an interactive visual question answering session with LLaVA that lasts at most $N$ rounds to extract salient information $z_t$ about object states and possible actions. We expand the observation at timestep $t$ to include both the current image $I_t$ and an image that is a small time interval after, $I_{t+\Delta t}$, i.e., $o_t = (I_t, I_{t+\Delta t})$. This gives GPT-4 the ability to extract more information to predict action from essentially before and after images. We adopt a ReAct (Yao et al., 2023) like framework. In each round, GPT-4 first reasons about the existing information, which includes its past prediction $(s_{t-1}, a_{t-1})$ and the information $z_t$ that has extracted so far, represented as a chat log between GPT-4 and LLaVA. Then, it can decide to either ask questions about one of

the images or end the conversation early. Meanwhile, LLaVA, which is only capable of examining one image at a time, responds to GPT-4's questions in detail.

Fig. 2 shows an example of GPT-4's reasoning and interaction with LLaVA. During its reasoning, GPT-4 notices that this timestep has similar objects as before, so it infers that the current action could be a continuation from the previous timestep. Based on this, GPT-4 decides to specifically ask about the slow cooker and the plate in the current image $I_t$. Then, because LLaVA answers that "human is about to put food on the plate," GPT-4 decides to ask about the status of these items in the future image $I_{t+\Delta t}$ to gain a comprehensive understanding of the potential action being executed.

## 4.2 PHASE II: STATE-ACTION PREDICTION

In this phase, GPT-4 uses information $z_t$ extracted in Phase I to predict states and action for the current timestep. Specifically, GPT-4 is first asked to predict the states in the current image and future image $(s_t, s_{t+\Delta t})$ given information $z_t$ and its previous state-action prediction $(s_{t-1}, a_{t-1})$. Then, given $z_t$, $(s_{t-1}, a_{t-1})$, and $(s_t, s_{t+\Delta t})$, GPT-4 needs to predict the action $a_t$ that has happened at the current timestep. Before each prediction, GPT-4 is instructed to critically examine LLaVA's answers in $z_t$ and its past state-action predictions because LLaVA produces noisy output, and GPT-4 might make mistakes in its previous prediction. In addition, because GPT-4 needs to make open vocabulary prediction, we instruct it to define new states and actions as necessary after we provide some examples of the common ones (e.g. `pick_up(obj)`). Fig. 2 shows examples of domain-specific states and actions (e.g. `is_filled_with(obj1, obj2)`, `serve(tool, obj, loc)`) that GPT-4 has defined to suit the demonstration's context. Once GPT-4 finishes Phase I and predicts state-action in Phase II for all timesteps, the generated state-action sequence gets converted to a robot task code by using prior works (Liang et al., 2022; Wang et al., 2023).

## 5 EXPERIMENTS

### 5.1 EXPERIMENTAL SETUP

**Baselines** We compare against 2 baselines that use the Socratic Model (Zeng et al., 2022) framework where a LLM model (GPT-4) queries a VLM model according to a *fixed* interaction setting.

1. SM-CAPTION: First, for each timestep $t$, the VLM is asked to generate a descriptive caption about the current image $o_t = I_t$. Then, GPT-4 uses $k = 10$ timesteps' captions and its prediction for the previous 2 timesteps to predict $k$ state-action.

2. SM-FIXED-Q: Building upon SM-CAPTION, the VLM is given a series of hand-crafted questions, and its answers for each timestep are sent to GPT-4. We specify questions about: the positional relations between objects in the image, the physical status of the objects (e.g. open, closed), the relation between the objects and the person (e.g. holding, touching), and the action being performed. GPT-4 uses VLM's answers for $k = 10$ timesteps and its prediction for the previous 2 timesteps to predict $k$ state-action.

**Models** We use GPT-4 with temperature 1.0 and max_tokens = 2048. For the VLM, we use LLaVA (Liu et al., 2023). Specifically, we use their model that is initialized with LLaMA-2-7B-chat parameters (Touvron et al., 2023b) and fine-tuned for 1 epoch on 80K image-text pairs of instruction-following data. We run zero-shot inference on 1 A6000 GPU.

**Dataset** We focus on EPIC-Kitchens (Damen et al., 2018; 2022), an egocentric video dataset of humans performing daily chores in their kitchen. We evaluate on videos in the validation dataset that have object annotations from EPIC-VISOR (Darkhalil et al., 2022). This subset contains 16 videos with an average of 60 timesteps per video, and we annotated state-action sequences for each video. To cluster these videos, we define 5 activity categories ("Cook a Dish", "Make a Snack", "Make a Drink", "Organize", "Do the Dishes") and assign these to the videos based on the high-level video description provided by EPIC-Kitchens. Each video could belong to multiple categories.

**Metrics** Evaluating open-vocabulary prediction is difficult without human evaluators, but recent LLMs can imitate this evaluation quality (Chiang & yi Lee, 2023). We use `gpt-3.5-turbo` as

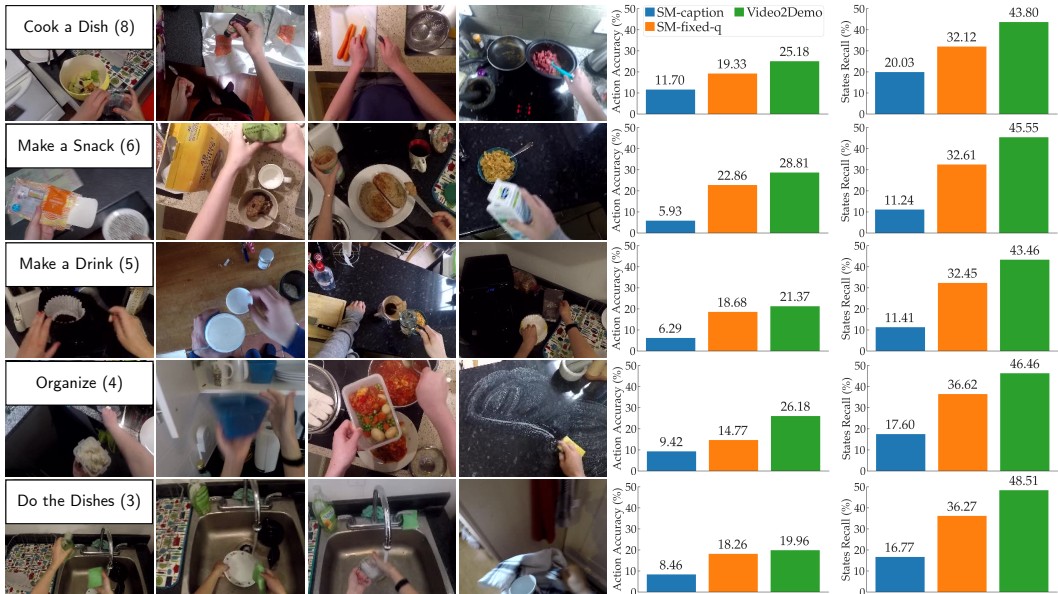

Figure 3: Average action accuracy and state recall grouped by activity categories. We report each category's number of videos. These videos each have a different kitchen background and an unique set of objects, so we show various example frames for each activity.

an indicator function $I_{\mathrm{GPT}}$ to determine whether a predicted action $\hat{a}$ or state $\hat{s}$ is semantically the same as an annotated action $a^*$ or state $s^*$. For a video with $T$ timesteps in total, we report the action accuracy and state recall. We do not evaluate state precision because we do not want to penalize the model for predicting a super-set of human-annotated states. We report two metrics:

$$\text{Action Accuracy} = \frac{\sum_{t=1}^{T} I_{\mathrm{GPT}}(\hat{a}_t, a_t^*)}{T}, \text{State Recall} = \frac{\sum_{t=1}^{T}(\sum_{p^* \in s_t^*} \min(\sum_{\hat{p} \in \hat{s}_t} I_{\mathrm{GPT}}(\hat{p}, p^*), 1))}{\sum_{t=1}^{T} |s_t^*|}$$

To evaluate code generation, we use the same code generation pipeline as (Wang et al., 2023). We first generate state-action sequences with our approach, then we use their pipeline to generate code and calculate ChrF score (Popović, 2015; Evtikhiev et al., 2023) against their reference code.

## 5.2 Results and Analysis

Table. 1 shows each method's overall performance for all EPIC-Kitchens' validation data that has state-action annotations. VIDEO2DEMO has the highest average action accuracy and state recall compared to the baselines. Between the two baselines, SM-FIXED-Q significantly outperforms SM-CAPTION, which means asking LLaVa specific questions rather than free-form captioning is the key to achieving high performance. We break down these results through series of questions to gain more insight about VIDEO2DEMO.

**How does performance vary across different categories of kitchen tasks?** Fig. 3 shows each method's average action accuracy and state recall for each activity category. VIDEO2DEMO consistently outperforms all baselines across all activity categories, with performance gains being more or less uniform. This demonstrates that VIDEO2DEMO generalizes on a wide range of kitchen tasks.

Table 1: Average action accuracy and state recall for all 16 videos.

|  | Action (%) | States (%) |
| --- | --- | --- |
| SM-CAPTION | 8.45 | 15.70 |
| SM-FIXED-Q | 19.60 | 33.11 |
| VIDEO2DEMO | 25.76 | 43.98 |

Interestingly, "Do the Dishes" category is the most challenging category for VIDEO2DEMO to predict action (19.96%), but the easiest one to predict states (48.51%). When users perform this type of task, they tend to stay at the sink and interact with a similar set of objects across multiple timesteps.

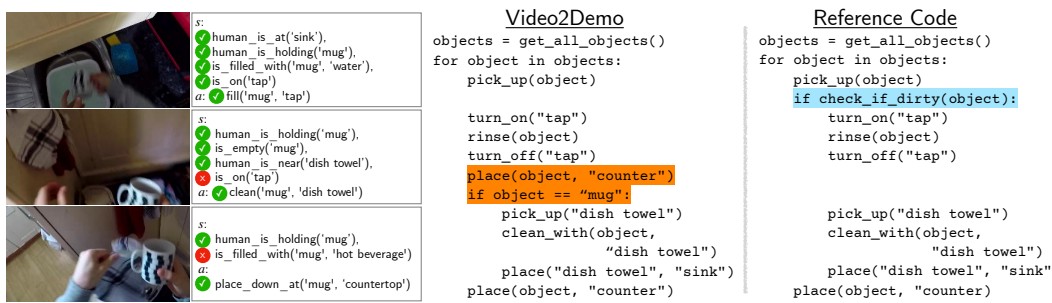

Figure 4: VIDEO2DEMO's state-action sequences can help generate code that resembles code written by experts. We label the correctness of the predicted states and action. For better visualization of the differences, we also added whitespace to align the code and highlight the differences between VIDEO2DEMO's code (in orange) and the reference code (in blue).

Table 2: Results for video segments that only show the dish-washing activity. We also include result of the code generated from manually annotated state-action demonstrations. We use ChrF (Popović, 2015), which is calculated by comparing the generated code against reference code written by an expert.

| | P30-07 | P22-05 | P04-101 | P07-10 | P22-07 | P30-08 | P7-04 | | Overall | |
| | | | | ChrF Scores | | | | ChrF | Action Acc | State Rec |
| --- | --- | --- | --- | --- | --- | --- | --- | --- | --- | --- |
| SM-CAPTION | 0.574 | 0.424 | 0.694 | 0.340 | 0.578 | 0.590 | 0.447 | 0.521 | 13.10 | 20.48 |
| SM-FIXED-Q | 0.619 | 0.520 | 0.882 | 0.638 | 0.622 | 0.540 | 0.500 | 0.617 | 18.31 | 35.47 |
| VIDEO2DEMO | 0.791 | 0.744 | 0.703 | 0.689 | 0.663 | 0.563 | 0.527 | 0.668 | 20.25 | 59.27 |
| ANNOTATED-DEMO* | 0.656 | 0.844 | 0.843 | 0.771 | 0.815 | 0.788 | 0.627 | 0.763 | - | - |

Because VIDEO2DEMO asks GPT-4 to consider its past predictions when it determines what questions to ask LLaVA, it is able to effectively use the continuity across timesteps to predict states well. However, because different dish washing actions that involve the same objects are visually similar, LLaVA is often unable to detect small details (e.g. whether a plate is dirty or has soap on it) that help distinguish actions like soaping the plate and rinsing the plate from each other.

**How well does each method perform on the final task of robot task code generation?** Our overall goal is to generate state-action sequences that can be used by downstream language model to output robot task code. We take state-action sequences predicted by each method and feed it to a pipeline from prior work (Wang et al., 2023) which accepts a text representation of state-action trajectories as input to generate code. They selected 7 videos of dish washing activity from EPIC-Kitchens dataset and trimmed each video to just contain the dish-washing segments. For each video segment, they manually annotated state-action sequences and defined ground-truth code that captures the user's dish washing behavior.

Table 2 show that, compared to our baselines, VIDEO2DEMO produces higher quality code that are more similar to the reference code. We also show a qualitative result in Fig. 4 that compares code generated from VIDEO2DEMO's state-action sequences with the reference code written by experts. Overall, VIDEO2DEMO's code captures most of the key actions (e.g. `rinse(object)`, `clean_with(object, 'dish towel')`. However, the code still differs in some control flow because (1) the VLM is not able to observe whether an object is dirty, and (2) the code generation model overfits to this specific demonstration, where the user has only washed a mug.

**How well does interactive question answering improve perform vs a fixed set of questions?** We provide a qualitative example in Fig. 5, illustrating how VIDEO2DEMO's ability to ask LLaVA questions based on existing conversation improves state-action prediction. At this timestep, the user is about to pick up to scissors to cut open the raisin bag. When LLaVA is prompted to freely describe the scene (SM-CAPTION), it hallucinates that the user is holding a spoon and a cup. Then, because GPT-4 simply translates LLaVA's caption, it can only generate states based on LLaVA's hallucinated description and is unable to determine the action due to the lack of concrete information.

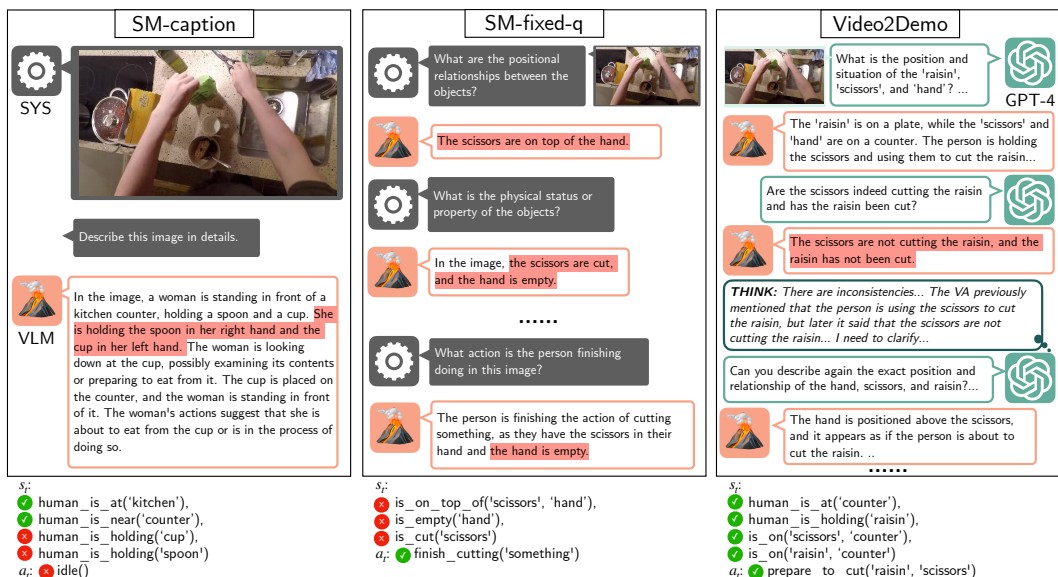

Figure 5: Qualitative example that compares VIDEO2DEMO's interactive question answering approach against baseline methods. We label the correctness for each approach's state-action prediction and highlight in red where LLaVA makes mistakes in its answers.

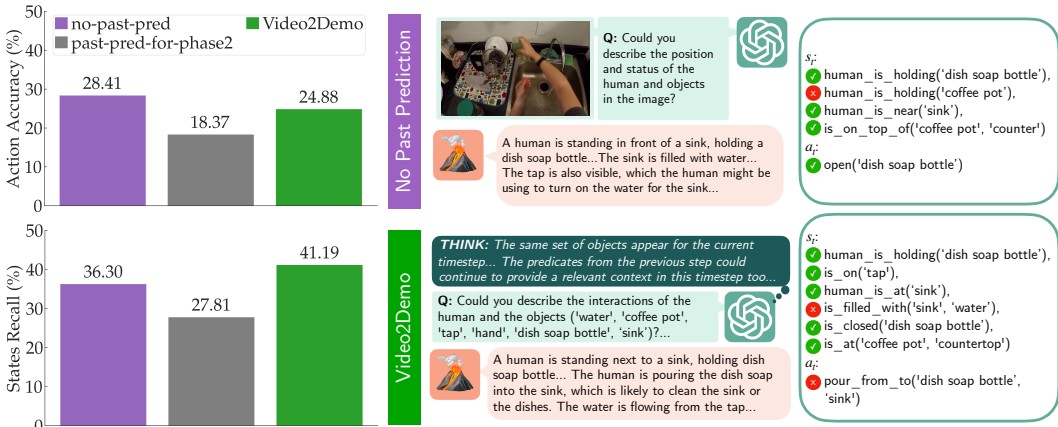

Figure 6: Ablation of having access to past prediction at different phases of VIDEO2DEMO's pipeline. We also show a qualitative example and illustrate the trade-off that past predictions improves states recall but worsens action accuracy.

SM-FIXED-Q successfully determines that the user is cutting something, but it struggles with state prediction because the fixed interaction template with hand-crafted questions are not able to ask LLaVA clarification questions. For example, LLaVA makes contradictory statements that "the scissors are on hand" and "hand is empty". Because GPT-4 cannot directly interact with LLaVA in this setting, it as a translator can only convert these contradictory statements into incorrect states.

In contrast, VIDEO2DEMO asks follow-up question about the exact interaction between the hand, scissors, and raisins after it notices inconsistency in LLaVA's previous responses. These clarifying questions help provide more details, thereby allowing VIDEO2DEMO to output the correct action "prepare_to_cut('raisin', 'scissors')" and more accurate states.

**How well does having access to past prediction affect the performance?** VIDEO2DEMO instructs GPT-4 to reason about information that is extracted so far and its previous state-action prediction. To evaluate the effect of past predictions, we define two ablations. NO-PAST-PRED excludes

past prediction during the entire process, while PAST-PRED-FOR-PHASE2 excludes it only during Phase I (sec. 4.1). We evaluate these approach on 5 videos, one from each kitchen activity category. Fig. 6 shows that although states and action in the previous timestep help VIDEO2DEMO to achieve higher state recall, this extra information tends to hinder its action accuracy. Meanwhile, PAST-PRED-FOR-PHASE2 is the worst performing approach, which suggests that past prediction is more useful in Phase I, where it can help extract information about the states.

VIDEO2DEMO, which has access to past predictions in Phase I, can begin the conversation by asking LLaVA about the current states of the objects that have appeared in the past. In Fig. 6's example, because VIDEO2DEMO notices that the current timestep has the same set of objects as the previous timestep, it asks LLaVA about the states of specific objects (e.g. "dish soap bottle", "sink", "tap"). In contrast, NO-PAST-PRED only asks a general question. The detailed questions help LLaVA to focus on specific parts of the image and provide more detailed information, thereby improving the states that VIDEO2DEMO eventually predicts.

However, these detailed questions sometimes also add bias to LLaVA's answer and cause it to self-contradict as it tries to mention all the objects that GPT-4 asks about. In Fig. 6, to answer VIDEO2DEMO's question, LLaVA describes that "the human is pouring the dish soap into the sink," which GPT-4 eventually uses as evidence to predict the incorrect action.

## 6 DISCUSSION

Grounding vision-language demonstrations in robot states and actions is critical for imitation learning from real world user data, and it largely remains as an open challenge. We leverage recent advances in both Vision-Language Models and reasoning capability of Large Language Models to tackle this problem. We propose a framework VIDEO2DEMO that enables GPT-4 to interactively query a VLM model to extract salient information from images and use this information to predict temporally consistent state-action sequences. We use a real-world kitchen activity dataset, EPIC-Kitchens (Damen et al., 2018; 2022), to evaluate our approach against baselines that used only fixed information exchange between GPT-4 and the VLM. We show that VIDEO2DEMO consistently outperforms baselines across a number of different real-world kitchen tasks.

**VIDEO2DEMO asks salient questions that results in responses with high signal-to-noise ratio.** Because VIDEO2DEMO instructs GPT-4 to reason about existing conversations before asking a question, GPT-4 asks salient, context-dependent questions. GPT-4 can also query the VLM with follow-up questions to gain more detail or clarification questions to resolve inconsistency.

**VIDEO2DEMO ensures that the sequence of predicted states and actions are temporally consistent.** GPT-4, by reasoning about past predictions, maintains the most plausible narrative about the underlying sequence of states. This helps it make correct predictions even when image degrades or the VLM model hallucinates.

## 7 LIMITATIONS

VIDEO2DEMO is limited by the capabilities of generative Vision-Language Models. Although GPT-4 can interactively query LLaVA and ask clarification questions to filter noises in LLaVA's answers, GPT-4 will not be able to resolve inconsistency if LLaVA continuously make incorrect statements about the image. LLaVA is not the only VLM model that faces hallucination issues. Many VLMs are plagued with common pitfalls that smaller or older LLM models face: hallucinations (Ji et al., 2023) due to bias in training data of image-text pairs, hash collisions in image embeddings (Tong et al., 2023), etc. Furthermore, EPIC-Kitchens dataset is challenging in many aspects: actions and key objects are often occluded; egocentric images are different from what the VLMs tend to be trained on; human's rapid movement causes motion blurs, etc. However, our framework is modular and invites several methods to tackle these issues. For example, GPT-4 can query an ensemble of VLM models and rely on answers that are consistent across most models. Another approach is to allow GPT-4 to probe the VLMs with different segments of an image, like what vision transformers do (Caron et al., 2021; Dosovitskiy et al., 2021) In addition, currently LLaVA can only examine one image at a time. Our framework can allow interactive querying between GPT-4 and a video language model that can reason about temporal relations between video frames.

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

Table 3: Results for video segments that only show the dish-washing activity. We also include result of the code generated from manually annotated state-action demonstrations. We compute ChrF (Popović, 2015) and CodeBERT Feng et al. (2020) by comparing the generated code against reference code written by an expert.

| | P30-07 | | P22-05 | | P04-101 | | P07-10 | | P22-07 | | P30-08 | | P7-04 | | Overall | |
|---|---|---|---|---|---|---|---|---|---|---|---|---|---|---|---|---|
| | ChrF | CodeBERT | ChrF | CodeBERT | ChrF | CodeBERT | ChrF | CodeBERT | ChrF | CodeBERT | ChrF | CodeBERT | ChrF | CodeBERT | ChrF | CodeBERT |
| SM-CAPTION | 0.574 | 0.839 | 0.424 | 0.787 | 0.694 | 0.897 | 0.340 | 0.811 | 0.578 | 0.809 | 0.590 | 0.879 | 0.447 | 0.902 | 0.521 | 0.846 |
| SM-FIXED-Q | 0.619 | 0.854 | 0.520 | 0.762 | 0.882 | 0.943 | 0.638 | 0.853 | 0.622 | 0.849 | 0.540 | 0.917 | 0.500 | 0.897 | 0.617 | 0.868 |
| VIDEO2DEMO | 0.791 | 0.956 | 0.744 | 0.866 | 0.703 | 0.852 | 0.689 | 0.910 | 0.663 | 0.822 | 0.563 | 0.847 | 0.527 | 0.912 | 0.668 | 0.881 |
| ANNOTATED-DEMO* | 0.656 | 0.851 | 0.844 | 0.975 | 0.843 | 0.911 | 0.771 | 0.961 | 0.815 | 0.965 | 0.788 | 0.942 | 0.627 | 0.908 | 0.763 | 0.930 |

Table 4: Average action accuracy and state recall for 2 videos.

| | Action (%) | | States (%) | |
|---|---|---|---|---|
| | P18_02 | P03_22 | P18_02 | P03_22 |
| SM-CAPTION | 8.00 | 4.20 | 9.70 | 8.00 |
| SM-FIXED-Q | 20.0 | 33.3 | 33.3 | 36.0 |
| VIDEO2DEMO | 24.0 | 33.3 | 50.0 | 56.0 |
| GPT4-V | 16.0 | 50.0 | 33.3 | 64.0 |

# APPENDIX

## A    ADDITIONAL EVALUATIONS

### A.1    CODEBERT EVALUATION ON DISHWASHING CODE

Table 2 shows the result of an downstream application of VIDEO2DEMO. Specially, we used VIDEO2DEMO to generate state-action sequences from real-world dish-washing videos in the EPIC-Kitchens dataset. Then, we use prior work Wang et al. (2023) to generate code that represents the high-level dish-washing task plan, corresponding to the state-action sequences. In Table 2, we compute ChrF scores Popović (2015) to compare the generated code against reference code written by an expert.

However, ChrF scores Popović (2015) is not the only metrics for evaluating code. Table 3 shows the CodeBERT score Feng et al. (2020) for comparing the generated code against the reference code. Overall, we observe that CodeBERT scores agree with ChrF scores, showing the code generated from VIDEO2DEMO's state-action trajectories are closer to the reference code compared to other baselines.

#### A.1.1    ADDITIONAL BASELINES USING GPT4-V

We conduct additional experiments using OpenAI's newly released GPT4-V (GPT-4-VISION-PREVIEW), which can accept images as input. We provide GPT4-V its past state-action predictions, the current image, and an image that is a small time interval after the current image. Then, we ask it to directly prediction the current state and actions. Due to current request limitations, we are able to compare our method to GPT4-V on two videos from EPIC-Kithens dataset, which is shown in Table 4. We observe that despite slight improvements in one demonstration (P03_22), GPT4-V is not immune to hallucinations and performs inconsistently across videos. However, it has promising results demonstrating its stronger visual understanding and instruction following ability compared to LLaVA. For future work, we will systematically evaluate GPT4-V on all videos, providing another baseline, and test Video2Demo's performance when GPT4-V is used as the VLM.

## B    QUALITATIVE ANALYSIS OF HOW VIDEO2DEMO'S PERFORMANCE VARY OVER TIMESTEP

In Fig. 7, we qualitatively examine how VIDEO2DEMO's action accuracy and states recall vary throughout time in an video by plotting the rolling average for both metrics with an window size of 4. This videos belong to "Make a Drink", "Make a Snack", and "Do the Dishes" categories

because its high-level video description in the EPIC-Kitchen's dataset is: "wash dishes, pour coffee and spread peanut butter on toast."

Fig. 8 provides insights on why both action accuracy and states recall have a noticeable drop at between timestep 10 to timestep 15. During this interval, the user is washing a clear coffee pot at the sink. Given the color of the coffee pot and the amount of less relevant objects around the sink, VIDEO2DEMO (and specifically LLaVA) struggles to determine the states of the coffee pot and what objects the user is interacting with. Visually, the clear coffee pot blends in with the silver sink, so VIDEO2DEMO consistently believes that the coffee pot is on top of the counter. Making false prediction about the coffee pot's positional states further causes most of the actions to not include coffee as one of the objects. Once the user finishes washing the coffee pot and moves on to making a peanut butter sandwich (starting from timestep 18), VIDEO2DEMO has better performance in predicting the action.

## C  EXAMPLE OF RUNNING VIDEO2DEMO ON ONE TIMESTEP

We provide the complete log of VIDEO2DEMO's Phase I and Phase II for the example frame $t$ that appears in fig. 1 and 2

### C.1  LOG OF PHASE I

In Phase I, GPT-4 interatively queries LLaVA various questions about the current image $I_t$ and the future image $I_{t+\Delta t}$. It eventually decides to finish asking the question. In the log below, the block that begins with "THINK" and has light green background are reasoning then questions that GPT-4 has in one round of conversation. LLaVA's response has light orange background, while system's response has light gray background.

```
THINK: I just got a new timestep to predict, so I should ask the system (
    SYS) to tell me what objects are in the scene before I ask the VA
    questions.
IMAGE: current image I_t
QUESTION: What objects are in this timestep?
```

```
SYSTEM: These objects are in this timestep: ['spoon', 'slow cooker', '
    plate', 'curry', 'hand']
```

```
THINK: The objects in the current timestep are the same as the objects in
     the previous timestep, so there might be some relationship or
    continuity between these two timesteps. From the rest of the context
    from the previous timestep, I can infer that the human was starting
    to serve food from the slow cooker onto a plate. It would make sense
    for the human to continue this action until the plate is filled. Thus
```

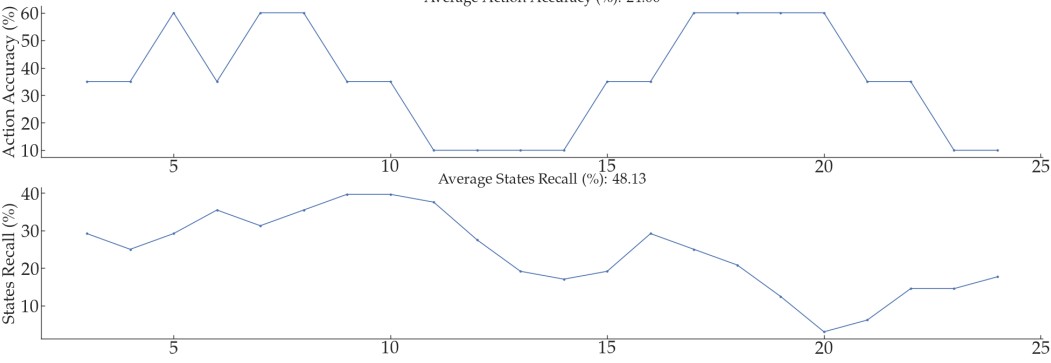

Figure 7: Rolling average (window of 4 timesteps) for action accuracy and state recall. The title of each plot shows the overall average for this video.

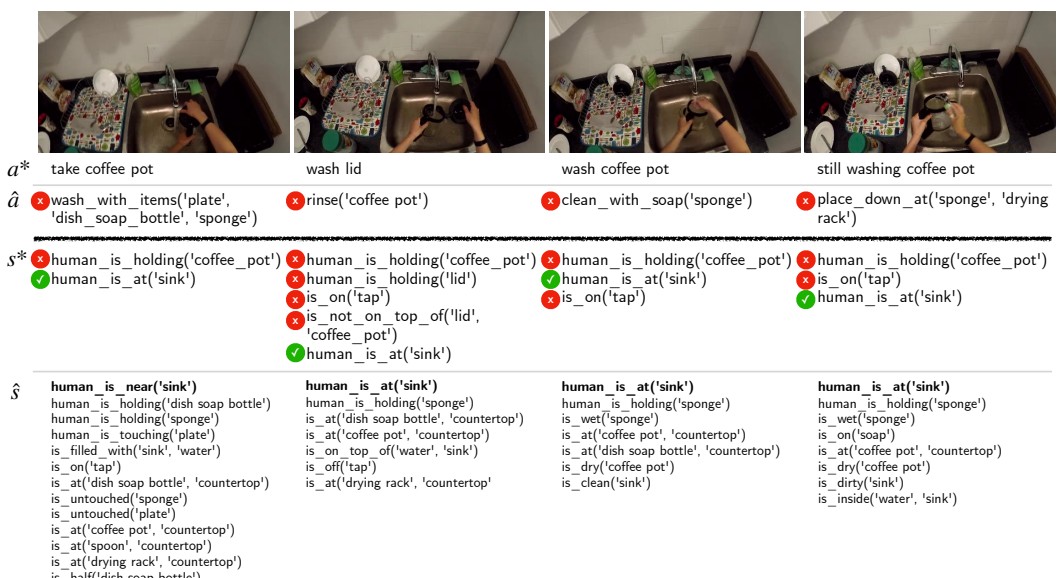

Figure 8: We show the current image $I_t$, predicted state-action $(\hat{s}_t, \hat{a}_t)$, and ground truth $(s_t^*, a_t^*)$ for four consecutive timesteps (11-14) that VIDEO2DEMO struggles to predict. We labels (1) the correctness of a predicted action and (2) whether each ground truth state predicate has a semantically predicate in the predicted states. VIDEO2DEMO fails to successfully identify the states of the coffee pot (and consequently the action about the coffee pot) because the coffee pot is clear and visually blends in with the sink in the background.

```
    , I should ask about the statuses of the slow cooker and the plate in
      the start image to verify this hypothesis. Because the image might
      also capture the human's action, I should ask about the relationship
      between the human and these objects as well.
IMAGE: current image I_t
QUESTION: Can you tell me about the statuses of the slow cooker and the
      plate, as well as the relationship between the human and these two
      objects?
```

```
LLAVA-ANSWER: The status of the slow cooker is that it is on the stove,
      and the status of the plate is that it is empty. The human is holding
       a spoon and is about to put food on the plate. The relationship
      between the human and these two objects is that the human is about to
       use the spoon to put food on the plate.
```

```
THINK: Given the existing context and objects in the image, it seems
      clear that the person is about to serve curry from the slow cooker
      onto the plate with the spoon. However, we still lack information
      about the end states. I will aim to investigate the locations of the
      objects, their statuses, and their relationships to each other and
      the human after the action has occurred to outline a comprehensive
      picture of what transpires in this timestep.
IMAGE: future image I_(t+delta t)
QUESTION: Could you please tell me about the locations and statuses of
      the slow cooker, plate, spoon, and curry in this image? Additionally,
       describe the human's interaction with these objects, as well as the
      positional relationship among these objects.
```

```
LLAVA-ANSWER: In the image, a human is holding a plate and a spoon while
      standing in front of a slow cooker. The slow cooker is filled with
      curry, and the human is about to eat the curry from the plate. The
      curry is located inside the slow cooker, and the human is holding the
       plate and spoon to serve themselves.
```

```
THINK: From the description of the VA, there are some inconsistencies.
   The VA says in the start image that the human is about to place food
   on the plate using a spoon, however, in the end image, the status of
   the curry is questioned for the end_image, the curry is in the slow
   cooker while the human is about to eat from the plate. But, according
    to our list of possible actions in this domain, consumption actions
   are not included. Thus, it's hard to infer that the human is about to
    eat curry just by looking at the images. There could be a
   misunderstanding or miscommunication here. To clarify the human's
   interaction and the condition of the plate and the curry in the end
   image, I need to ask the VA.
IMAGE: future image I_(t+delta t)
QUESTION: Does the plate still remain empty after the human's interaction
    with the slow cooker and spoon? Can you verify if the curry was
   transferred from the slow cooker to the plate?
```

```
LLAVA-ANSWER: Yes, the plate remains empty after the human's interaction
   with the slow cooker and spoon. The human is holding a spoon and has
   not yet transferred the curry from the slow cooker to the plate.
```

```
THINK: Based on the previous states and the visual assistant's responses,
    the human was preparing to serve food from the slow cooker onto the
   plate. Now, according to the visual assistant, the human is still
   holding the plate and spoon, but the curry hasn't been transferred
   yet. While there doesn't seem to be any major changes or actions, I
   want to make sure that no subtle changes have occurred either.
   Therefore, I need to confirm details about the slow cooker and the
   spoon, like their statuses, to see if any interactions have taken
   place that could signify an action.
IMAGE: future image I_(t+delta t)
QUESTION: Please describe the statuses of the slow cooker and the spoon.
   Also, could you tell me whether the human has interacted with either
   the slow cooker or the spoon in any way besides holding them?
```

```
LLAVA-ANSWER: The human is holding a spoon and a plate in their hand, and
    they are standing next to a slow cooker. The human has not
   interacted with the slow cooker or the spoon in any way, as they are
   simply holding them. The plate is also empty, indicating that the
   human has not eaten from it yet.
```

```
THINK: Given that I have no rounds left, I need to make my decision on
   the states and actions for this timestep. Looking at the queries and
   responses so far, the human was initially holding the plate and spoon
    near the slow cooker which was on, with curry inside it. By the end,
    the human was still holding the plate and spoon, had not interacted
   further with the slow cooker or spoon, and the plate was still empty
   indicating no curry was transferred. Hence, it would be appropriate
   to finish asking questions and proceed to state and action
   identification.
DECISION: finish_aksing_question
```

## C.2 LOG OF PHASE 2

In Phase II, GPT-4 first predicts states for the current image $I_t$ and the future image $I_{t+\Delta t}$ given extracted information (represented as chat log) from Phase I and its previous predictions. Then, it predicts the action.

```
THINK: Based on the VA's responses, it is evident that the slow cooker is
    on the stove and filled with curry, while the human is holding an
   empty plate and a spoon. The person is positioned in proximity to the
    slow cooker, preparing to serve themselves some curry. Although the
   action in the previous timestep suggests that the individual is about
    to serve the food onto the plate from the slow cooker, there is no
```

```
    actual confirmation of this action taking place. The VA reiterated
    multiple times that the plate remains empty, implying that the dish
    was not transferred onto it yet. Consequently, the start and end
    states of this timestep do not reflect a change in the condition of
    the plate.
CURRENT STATES PREDICTION:
[
    "human_is_holding('plate')",
    "human_is_holding('spoon')",
    "is_on('slow_cooker')",
    "is_filled_with('slow_cooker', 'curry')",
    "is_empty('plate')",
    "human_is_near('slow_cooker')"
]
FUTURE STATES PREDICTION:
[
    "human_is_holding('plate')",
    "human_is_holding('spoon')",
    "is_on('slow_cooker')",
    "is_filled_with('slow_cooker', 'curry')",
    "is_empty('plate')",
    "human_is_near('slow_cooker')"
]
```

```
THINK: After comparing the start and end states of this timestep, I
    notice there is no change in states, which suggests that the action
    carried out in this timestep might be related to holding objects in
    preparation for a subsequent action, rather than changing the states
    of the objects themselves. In the chat logs, the VA confirms that the
     person is holding a spoon and a plate, standing near a slow cooker
    filled with curry, but has not moved or transferred the curry on the
    plate. Considering all these aspects, it seems like the person is
    about to serve the curry from the slow cooker onto the plate, but the
     action has not been performed yet. This suggests the action in this
    timestep is 'prepare_to_serve_from'.
ACTION PREDICTION: prepare_to_serve_from('curry', 'slow_cooker')
```

# D   VIDEO2DEMO PROMPTS

## D.1   GPT-4'S PROMPT

### D.1.1   SYSTEM MESSAGE

```
[[Your goal]]
Your goal is to get information from several images that represent a
    timestep and determine (1) the action of that timestep, (2) the
    states right before that action, and (3) the states right after that
    action. You cannot directly see the images, but you can ask a visual
    assistant (VA) who can look at only one image at a time and answer
    your questions.
[[Information about the input]]
You will receive:
- [[Previous Timesteps]], which are the JSONs for the previous timesteps
    that you have predicted action and states for.
- [[Chat Log]], which is a log of all the conversations that you have had
     with the visual assistant for this timestep.
- [[Current Instruction]], which contains rules about what you can do at
    the current round.
[[General rules]]
Each time, you must first provide reasoning before choosing what to do.
    You must reply following this template:
[start_of_template]
{
 "reason": "<put_your_reasoning_here>",
 "act": "<put_what_you_are_doing_here>"
}
[end_of_template]
[[Domain]]
A person is doing some household chores.

State predicates to describe a timestep could include:
1) the location of the human (e.g. "human_is_at(<LOC>)", "human_is_near(<
    LOC>)", etc)
2) the relationship between the human and the objects (e.g. "
    human_is_holding(<A>)", "human_is_touching(<A>), "
    human_is_reaching_inside(<A>)", etc)
3) the positional relationship of the objects (e.g. "is_on_top_of(<A>, )", "is_underneath(<A>, )", "is_left_of(<A>, )", "is_right_of
    (<A>, )", "is_in_front_of(<A>, )", "is_behind(<A>, )", "
    is_inside(<A>, )", etc)
4) the status/property of the objects (e.g. "is_on(<A>)", "is_off(<A>)",
    "is_open(<A>)", "is_closed(<A>)", "is_empty(<A>)", "is_filled_with(<A
    >, )", "is_cut(<A>)", "is_cooked(<A>)", etc)
5) the location of the objects (e.g. "is_at(<A>, <LOC>)", "is_near(<A>, <
    LOC>)", etc)

You must assume that the person can only do one action at a time. Some
    examples of action predicates are: "pick_up(<A>)", "pick up_from(<A>,
     <location_B>)", "place_down(<A>)", "place_down_at(<A>, <location_B>)
    ", "open(<A>)", "close(<A>)", "turn_on(<A>)", "turn_off()", "wash
    (<A>)", "cut(<A>)", "cook(<A>)", "stir(<A>)", etc.

You must follow these rules when you are writing state predicates and
    action predicates:
- State predicates and action predicates should be represented as a
    Python function call.
- The function parameters should not contain strings that directly refer
    to the human (e.g. "person", "human", "user", "man", "woman", etc.)
    For example, if a person is near a sink, you should not say "is_near
    ('person', sink)". Instead, you should say "human_is_near('sink').
    Similarly, if a person's hand is touching a cup, you should not say "
```

```
    is_touching('hand', 'cup')". You should say "human_is_touching('hand
        ', 'cup')".
- You can use the predicates in the examples above, but you can also
    create new ones as long as they are defined like a Python function
    call.
```

### D.1.2   QUERY MESSAGE - PHASE I

`<previous_to_fill>` is replaced by state-action predictions from the previous timestep and `<chat_log_to_fill>` is replaced by current timestep's chatlog.

```
[[Previous State-Actions]]
<previous_to_fill>
[[Chat Log]]
<chat_log_to_fill>
[[Current Instruction]]
Currently, you are in the process of asking the visual assistant question
    . You have <rounds_left> rounds left, so you should ask questions
    that maximize the information gain.

You must follow these rules when you are deciding what to do:
## General information
- Given 2 images ('start_image' and 'end_image') of a timestep, Your goal
     is to ask questions that collect as much information as possible and
     help you determine the start states (representing the beginning of
    this timestep'), the end states (representing the end of this
    timestep), and the action that happened during the timestep.
## Rules for asking questions
- You should at least ask one question about "start_image" and one
    question about "end_image".
- You should not ask yes or no questions. You should not ask a question
    just about one state of one object in the image unless you really
    need to clarify an inconsistency.
- You should either ask about (1) one category of states for multiple
    objects or (2) multiple categories of states for one object. You
    should make the VA answer with as many details as possible. The VA
    should not be able to answer your question with just one sentence.
- The VA can only look at one image at a time, and it does not have any
    memory of its previous answers. The VA does not understand what it
    means for an image to be a "start_image" or an "end_image".
- Thus, you must not ask any questions that require the VA to compare two
     images. You must not ask it to compare an image with its previous
    answer or answer your question based on what image it has seen before
    . You must not ask it about changes: e.g. "what changes it see", or "
    how does something or the surrounding change", etc
## Rules for reasoning about the VA's answers:
- The VA tends to make mistakes, so you must critically analyze its
    answer.
- You must ignore any objects in VA's answers that are not in the initial
     object list. You should not ask questions about objects that are not
     in the object list.
- You should only ask clarification questions if the VA is inconsistent
    about objects in the object list.
## General work flow
You should follow this workflow when you think about what to ask:
1. First, you should compare the current timestep's objects with the
    previous timestep's objects.
2. If there is a high overlap in objects, you should ask the VA questions
     based on the context (the predicted states and action) from the
    previous timestep. Remember that the predicted states and action
    could be wrong. However, this timestep might have the same or a
    follow-up action from the previous timestep.
3. If there is not much overlap, you do not have much context information
    , so you need to start from scratch. You should ask questions that
```

```
     help you understand 1) the location of the human, 2) the relationship
      between the human and the objects, 3) the positional relationship of
      the objects, 4) the property/status of the objects, 5) the location
     of the objects, etc.

You can either (1) ask questions about one image ("start_image" or "
    end_image") or (2) decide to finish asking questions early. Questions
     could be in multiple sentences, but all sentences should be enclosed
     in the same string. Remember: you are not allowed to ask the VA to
    compare two images, or compare the current image with its previous
    answers, or answer what has changed. You must choose from one of the
    following templates:
- To ask a question about the start image:
{
 "reason": "<put_your_reasoning_here>",
 "act": {
        "image_to_ask_about": "start_image",
        "question": "<put_your_question_here>"
}
}
- To ask a question about the end image:
{
 "reason": "<put_your_reasoning_here>",
 "act": {
        "image_to_ask_about": "end_image",
        "question": "<put_your_question_here>"
}
}
- To finish asking questions early:
{
 "reason": "<put_your_reasoning_here>",
 "act": "finish_asking_question"
}
```

### D.1.3 QUERY MESSAGE - PHASE II

Once again, `<previous_to_fill>` is replaced by state-action predictions from the previous timestep and `<chat_log_to_fill>` is replaced by current timestep's chatlog.

```
[[Previous State-Actions]]
<previous_to_fill>
[[Chat Log]]
<chat_log_to_fill>
[[Current Instruction]]
Now, you must determine the start states and end states in this timestep.
     You need to critically examine the information from the chat log and
     the previous timestep, which could contain false information.

You must follow these rules when you are reasoning:
- You can trust the list of objects that are in the previous and current
    timesteps.
- You should critically examine the previous timestep's predicted states
    and action and the VA's answers because they could be incorrect. In
    each round of the conversation, the VA does not have any memory of
    its previous answers. Its answer is only based on looking at one
    image.

You must follow these rules when writing state predicates:
- You must produce a list of states for start states and end states. Each
     list needs to have at least one predicate.
- Each state predicate must be represented as a Python function call. The
     function parameters should be objects in the images (from the object
     list or the chat log).
```

```
- You can use the examples of state predicates in [[Domain]], but you can
    also define new ones.
- You must make your prediction following this format:
<start_of_template>
{
    "reason": "<put_your_reasoning_here>",
    "act": {
    "start_states": ["<state_predicate_1>", "<state_predicate_2>", ...],
    "end_states": ["<state_predicate_a>", "<state_predicate_b>", ...],
    }
}
<end_of_template>

action_prediction_instruction
Now, you must determine the action in this timestep. You can use the
    state states and end states that you have determined for this
    timestep. You need to critically examine the information from the
    chat log and the previous timestep, which could contain false
    information.

You must follow these rules when you are reasoning:
- You can trust the list of objects that are in the previous and current
    timesteps.
- You should critically examine the previous timestep's predicted states
    and action and the VA's answers because they could be incorrect. In
    each round of the conversation, the VA does not have any memory of
    its previous answers. Its answer is only based on looking at one
    image.
- You should also use the start states and end states that you just
    predicted in [[Current Timestep's States Prediction]]

You must follow these rules when writing action predicates:
- You must predict exactly one action. You cannot predict more than one
    action. You cannot predict that the human is doing nothing, and you
    cannot set the action to be "None", "null", "none", etc. The action
    should describe what the human is doing in the current timestep.
- An action predicate must be represented as a Python function call. The
    function parameters should be objects from this object list: <
    obj_list>
- You can use the examples of action predicates in [[Domain]], but you
    can also define new ones.
- You must make your prediction following this format:
<start_of_template>
{
"reason": "<put_your_reasoning_here>",
"act": {
"action": "<action_predicate>",
}
}
<end_of_template>
```

## D.2 CHAT LOG TEMPLATE

The chat log within `<chat_log_to_fill>` is formatted as:

```
REASONING: <reasoning>
IMG: <image-type>
Q: <question>
VA: <answer>
...
```

It is initialized at the start as:

```
REASONING: I just got a new timestep to predict, so I should first know
    what objects are in the scene.
```

```
IMG: start_image

Q: What objects to do you see in this timestep?
A: It's highly likely that I saw these objects: <obj_list>.
```

where `<obj_list>` is replaced with the object list in that timestep.

### D.3 LLAVA'S PROMPT

Since the question per image is now-controlled by GPT-4, the human just decides the system message

```
You are a helpful language and vision assistant. Each time, the user
    will provide a first-person, egocentric image and a list of
    objects that are highly likely to appear in the image. Not all
    objects might show up in the image because they could be occluded
    . If you don't see an object from the object list, you should be
    honest and say that you haven't seen that object. You must not
    describe any objects that are not listed. In your responses, you
    must answer the question that the human has specific to this
    scene and the objects listed.
```

## E  BASELINE PROMPTS

### E.1  GPT-4

GPT-4 acts as a translator from descriptions of the scene to state-action predicates, and so the prompts are common for both baselines.

#### E.1.1  SYSTEM MESSAGE

```
You are a helpful assistant who translates a list of scene descriptions
    into a list of state-action JSONs that are more well-formatted.

First, the user will provide the [[Domain]], which describes the general
    theme of the tasks and the rules about the state predicates and
    actions that are available to describe the scenes. Then, the user
    will provide [[Previous State-Actions]], which are state-action JSONs
     for previous timesteps that you have translated, and a list of [[
    Scene Descriptions]] that describe what happens in each timestep of
    someone doing some tasks in the domain's environment.

Your response should be a list of jsons. For each timestep {i} in the [[
    Scene Descriptions List]], you must follow the JSON template below
    and translate the timestep into multiple state predicates and one
    action:
<start of template>
[
{
        "timestep": ,
        "states": [<state_predicate_1>, <state_predicate_2>,...],
        "action": <action>
},...
]
<end of template>

You must follow these rules when you translate scene descriptions:
- You must replace and fill in anything within a sharp bracket <> in the
    template.
- When translating one timestep, you must consider [[Previous State-
    Actions]] and any state-action JSONs that you have just generated.
```

```
    The objects that appear in previous timesteps could still exist in
       the current timestep, and the past actions could affect the past
       states and cause the current states.
- The state predicates and actions must be defined like a Python function
       call. For example, a state predicate could be "at('cup', table')",
       and an action could be "place_at('cup, 'table')".
- In [[Domain]], the user will provide some examples of state predicates
       and actions that you can use. You can use those provided ones, but
       you can also create new ones as long as they are defined like a
       Python function call.
- A state-action JSON for one timestep can have multiple state predicates
       , but it can only have one action.
- You should ground each state predicate and action with things that are
       mentioned in the scene description. You must label them by their type
        (e.g. 'cup', 'table').
- You must not mention "person" or any words to refer to the human in the
        video. For example, if a person is holding a cup in the video, you
       should not say "is_holding('person', 'cup')". You should say "
       human_is_holding('cup')". You should never write "person", "human", "
       user", "man", "woman", etc.
<domain_definition>
[[Domain]]
A person is doing some household chores.

State predicates should be represented as a Python function call that
       will return True or False. State predicates should not contain the
       human as one of the function parameters. State predicates to describe
        a timestep could include:
1) the location of the human (e.g. "human_is_at(<LOC>)", "human_is_near(<
       LOC>)", etc)
2) the relationship between the human and the objects (e.g. "
       human_is_holding(<A>)", "human_is_touching(<A>), "
       human_is_reaching_inside(<A>)", etc)
3) the positional relationship of the objects (e.g. "is_on_top_of(<A>, )", "is_underneath(<A>, )", "is_left_of(<A>, )", "is_right_of
       (<A>, )", "is_in_front_of(<A>, )", "is_behind(<A>, )", "
       is_inside(<A>, )", etc)
4) the status/property of the objects (e.g. "is_on(<A>)", "is_off(<A>)",
       "is_open(<A>)", "is_closed(<A>)", "is_empty(<A>)", "is_filled_with(<A
       >, )", "is_cut(<A>)", "is_cooked(<A>)", etc)
5) the location of the objects (e.g. "is_at(<A>, <LOC>)", "is_near(<A>, <
       LOC>)", etc)

You must assume that the person can only do one action at a time. The
       action may or may not involve objects or locations in the environment
       . Action should not contain the human as one of the parameters. Some
       examples of actions are: "pick_up(<A>)", "pick up_from(<A>, <
       location_B>)", "place_down(<A>)", "place_down_at(<A>, <location_B>)",
        "turn_on(<A>)", "turn_off()", "wash(<A>)", "cut(<A>)", "cook(<A>)
       ", "stir(<A>)".

You can also define new state predicates and actions as long as they are
       written as a Python function call,  they don't overlap with the
       existing ones, and they don't explicitly mention the human as one of
       the function parameters.

You can assume that there are no significant changes from one timestep to
        another. This means that objects in the current timestep are likely
        to be the same as the ones mentioned in the previous timestep.
```

### E.1.2  QUERY MESSAGE

This contains the answers from LLaVA that GPT-4 needs to translate into state-action sequences. `<previous_to_fill>` is replaced with the last two of its predictions, and `<all_descriptions_to_fill>` contains $k = 10$ of LLaVA's answers.

```
[[Previous State-Actions]]
<previous_to_fill>
[[Scene Descriptions]]
<all_descriptions_to_fill>
```

### E.2  LLAVA'S PROMPT

### E.2.1  SM-CAPTION

For SM-CAPTION, we intend to use LLaVA as is to gauge how well it performs, and use GPT-4 as a translator to state-actions sequences from descriptions.

**System message**  This is the default provided in Liu et al. (2023)

```
You are a helpful language and vision assistant. Each time, the user
    will provide an image. Your goal is to describe the scene with as
     many details as possible. Your description should be truthful,
    helpful, and detailed.
```

**Question per image**  We tailor this question to maximize details within one question.

```
Please describe in detail what states and action are happening in
    this image. The states could be about: 1) the location of the
    human, 2) the relationship between the human and the objects, 3)
    the positional relationship of the objects, 4) the property/
    status of the objects, 5) the location of the objects, 6) the
    action of the human, etc.'
```

### E.3  SM-FIXED-Q

### E.3.1  LLAVA'S PROMPT

**System message**  For SM-FIXED-Q, we add additional context.

```
You are a helpful language and vision assistant. Each time, the user
    will provide a first-person, egocentric image and a list of
    objects that are highly likely to appear in the image. Not all
    objects might show up in the image because they could be occluded
    . You should not describe any objects that are not listed. In
    your responses, you must only mention objects in the initial
    object list provided by the user.  Your goal is to describe the
    scene using these objects and provide helpful, detailed, specific
     answers to the user's questions.
```

**Questions per image**  We ask the following questions in a QA style to elicit specific answers from LLaVA. `<obj_list>` is replaced by objects per timestep.

```
Q1: The list of objects that might be in this image is: Mobj_list>.
    What are the positional relationships between only these objects?
     For example: "A is on top of B", "A is to the left of B", "A is
    to the right of B", "A is in front of B", "A is behind B", "A is
    inside B". You can choose from the examples or define new
    positional relationships.'
```

Q2: Only using the objects from the list in the first question, what
    is the physical status or property of these objects? For example:
     "A is cut", "A is cooked", "A is open", "A is closed", "A is
    empty", etc. You can choose from the examples or define new ones.

Q3: Only using the objects from the list in the first question, what
    is the relationship between the person and those objects? For
    example, "is the person holding something", "is the person using
    something", etc. You can choose from the examples or define new
    relationships.

Q4: This image is at the end of when the person does an action. What
    action is the person finishing doing in this image? You must not
    answer what the person might do next. You should infer and answer
     what the person is about to finish doing or just finished doing.
     Some actions could be: "picking up something", "placing down
    something", "turn on something", "turn off something", "wash
    something", "cut something", "cook something", "stir something",
    etc. You can choose from these example actions or define new
    actions.

