# OpenReview forum: "Video2Demo: Grounding Videos in State-Action Demonstrations"
_ICLR.cc/2024/Conference — Submitted to ICLR 2024_

### Official Review · Reviewer_RP75 · 2023-10-29

**Soundness:** 3 good
**Presentation:** 3 good
**Contribution:** 2 fair
**Rating:** 5
**Confidence:** 4

**Summary:**

The paper proposes to use GPT-4 to interactively query a VLM to construct temporally coherent state-action sequences. Then it uses a prior method, Demo2Code, to generate robot task code that faithfully imitates the demonstration. Experiments on EPIC-Kitchens show it outperforms prior VLM-based approaches.

**Strengths:**

* The paper proposes an effective way to convert human video demonstrations to state-action sequences, which are useful for generating executable robot policies.
* The paper conducts extensive experiments on EPIC-Kitchens.
* The paper is well-written and easy to follow.

**Weaknesses:**

* **Other LLMs and VLMs**: How do other LLMs and VLMs perform on this task? I am curious to see how this framework is generalized to other models.
* **GPT4-V**: It would be good to include some results of GPT-4V. I know its API is not released yet, but some quick experiments through ChatGPT's UI are sufficient.
* **Execution-based evaluation**: I wonder whether you can provide some execution-based results of robot code to prove the generated state-action sequences are really useful.
* **Prior works**: It would be good to discuss the paper's relationship to some additional prior works:

[1] ProgPrompt: Generating Situated Robot Task Plans using Large Language Models. Singh et al.

[2] VoxPoser: Composable 3D Value Maps for Robotic Manipulation with Language Models. Huang et al.

[3] Voyager: An Open-Ended Embodied Agent with Large Language Models. Wang et al.

**Questions:**

See weakness.

---

> ### Author Response · Authors · 2023-11-16
>
> We thank the reviewer for their feedback, the comparison with newer vision model APIs and appreciate the questions raised about the the evaluation of our pipeline
>
> ## **Questions and Weaknesses**
> ### **Clarification on Prior Work:**
> We thank the reviewer for their suggestion to consider ProgPrompt[5], Voyager[6] and Voxposer[7] as related works. In our next revision, we will duly acknowledge them.
>
> We wish to emphasize that `Video2Demo` serves a different purpose from the mentioned works.
>
> 1. ProgPrompt is an LLM-based task planner translating language instructions to code
> 2. Voxposer defines affordances and rewards through code for motion planning
> 3. Voyager builds a skill library for playing Minecraft using LLMs.
>
> `Video2Demo` however, is not a planner. We focus on extracting state-action predicates through vision-language demonstrations. We use Demo2Code[8] as an example of how `Video2Demo`’s output can be used in downstream task planners like [5-7]. We aim to explore this in future work with more embodied planners.
>
>
> ### **Other VLMs and LLMs:**
> Our choice for the generative LLM was straightforward: there are currently no other LLMs that match the context capacity, instruction following and logical coherence capabilities that GPT-4 has [1]
>
> Our generative VLM choice was based on recent literature surveys, underlying simplicity of the architecture, and the choice of pre-trained models in LLaVA [2,3]. Due to time and cost limitations, we could not ablate the choice of the VLMs. However, the method is plug and play for both the choice of LLM and VLM, similar to the Socratic Model approach.
>
> Finally, OpenAI has recently enabled GPT4-vision as part of their API[4]. For future work, we are interested in evaluating this in our setting as:
> 1. An alternative generative VLM in the `Video2Demo` pipeline, where we can expect better visual grounding compared to LLaVA.
> 2. A competing method to `Video2Demo`
>
> ### **On execution-based evaluation**
> In the global response, we elaborate on how we primarily focus on the problem of extracting rich information from offline vision-language demonstrations. In future work, we plan to evaluate `Video2Demo`’s generated state-action with task and motion planners [5,11] in simulators [9-10].
>
>
>
> ### References -
>
> [1] OpenAI, R. "GPT-4 technical report." arXiv (2023): 2303-08774.
>
> [2] Liu, Haotian, et al. "Visual instruction tuning." arXiv preprint arXiv:2304.08485 (2023).
>
> [3] Touvron, Hugo, et al. "Llama 2: Open foundation and fine-tuned chat models." arXiv preprint arXiv:2307.09288 (2023).
>
> [4] OpenAI: New models and developer products announced at DevDay. https://openai.com/blog/new-models-and-developer-products-announced-at-devday, 2023
>
> [5] Singh, Ishika, et al. "Progprompt: Generating situated robot task plans using large language models." 2023 IEEE International Conference on Robotics and Automation (ICRA). IEEE, 2023.
>
> [6] Huang, Wenlong, et al. "Voxposer: Composable 3d value maps for robotic manipulation with language models." arXiv preprint arXiv:2307.05973 (2023).
>
> [7] Wang, Guanzhi, et al. "Voyager: An open-ended embodied agent with large language models." arXiv preprint arXiv:2305.16291 (2023).
>
> [8] Wang, Huaxiaoyue, et al. "Demo2Code: From Summarizing Demonstrations to Synthesizing Code via Extended Chain-of-Thought." arXiv preprint arXiv:2305.16744 (2023).
>
> [9] Eric Kolve et al. AI2-THOR: An Interactive 3D Environment for Visual AI. arXiv:1712.05474, 2022
>
> [10] Todorov, Emanuel, Tom Erez, and Yuval Tassa. "Mujoco: A physics engine for model-based control." 2012 IEEE/RSJ international conference on intelligent robots and systems. IEEE, 2012.
>
> [11] Yu, Wenhao, et al. "Language to Rewards for Robotic Skill Synthesis." arXiv:2306.08647, 2023

---

> > ### Author Response · Authors · 2023-11-21
> > **Thank you for your review**
> >
> > Hello!
> >
> > Thank you again for your feedback and questions, and suggestions to strengthen our paper!
> > As per your suggestion, we have added a qualitative comparison against GPT4-V on the same task, in the global response.
> >
> > As the discussion period ends soon, please let us know if we can provide any additional clarifications or answers that would help you in your evaluation.

---

> > > ### Comment · Reviewer_RP75 · 2023-11-21
> > >
> > > Thanks a lot for the rebuttal! I am still concerned with the lack of execution-based evaluation, which is also a universal concern among all the reviewers. I really hope to see systematic experiments in either a simulator or a real robot.

---

### Official Review · Reviewer_51er · 2023-11-01

**Soundness:** 3 good
**Presentation:** 3 good
**Contribution:** 2 fair
**Rating:** 5
**Confidence:** 3

**Summary:**

The paper "Video2Demo" addresses the challenge of teaching robots everyday tasks through a novel approach using a combination of GPT-4 and Vision-Language Models (VLMs) like LLaVA. The approach consists of 2 phases. First, GPT-4 is prompted to  interact with the VLM to create temporally coherent state-action sequences from descriptive VLM captions. Moreover, GPT-4's capacity to follow up with the VLM for additional information or clarification further enhances the quality of the obtained responses. Second,  GPT-4 is prompted to generate robot task code that imitates the demonstrated actions. The approach is evaluated on the EPIC-Kitchens dataset, outperforming other methods. Key contributions include a new framework for transforming vision-language demonstrations into state-action sequences, annotated data for benchmarking, and superior performance in both state-action and code generation tasks.

**Strengths:**

Originality:  Video task description requires a combination of object identification, contextual analysis over time, and the application of common knowledge and reasoning to provide a comprehensive and coherent account of the video's content and events.
The paper introduces a useful system design with prompt engineering for interactive dialog between Vision-Language Models (VLMs) and Language Models (LLMs), providing a fresh perspective on task planning based on video data.

Quality:  The paper frames the problem of decoding what is happening in a video in the form of iterative dialog between VLM (like LLaVA) to answer queries about a frame and LLM (like GPT-4) for asking questions, and  deciding state-action predicates.
The research is validated on real-world data (EPIC-Kitchens), and outperforms baselines in both state-action and code generation.

Clarity: The paper is well-structured and accessible, making it easy for a wide audience to understand. The key research questions and the failure cases are well discussed. The paper brings the problem of spatial grounding and hallucinations in LLMs and VLMs to the community's attention. This is reflected in the lower accuracy in symbolic state recall and action prediction.

Significance: The paper addresses a significant challenge in robotics and AI, with potential applications in various domains, and introduces a valuable approach for interactive AI systems.

**Weaknesses:**

1. The paper motivates the problem of "teach robots everyday tasks". But there are no simulated or real robots experiments which makes it hard to assess the practicality of proposed approach and the possible failure scenarios. For example, how would the generated task plan compare to execute task in simulated environments like ALFRED [1].  The scope and possible future implication can be clear, like the proposed solution seems well suited for video comprehension, that can facilitate task planning.
1. One of the reasons why the proposed approach may be unsuitable for robot is the possibility of compounding error over interactive dialog and the corresponding latency.
1. Video2Demo relies heavily on prompt engineering, which requires considerable effort. It is unclear if the presented prompts are applicable to just EPIC kitchen videos only, or can be applied more broadly to other activity videos.

**Questions:**

1. Can it scale to beyond egocentric videos in EPIC kitchen to third-person tutorial videos? How would the prompt change, especially in the phase 2 where the prompts and state-action predicates seems to be centered on the human in the videos?
1. How does ChrF compare to other code generation metrics [1]? Does the generated code with high ChrF score correlate with human preference? How much of the generated code follow required syntax and physical feasibility to execute successfully on a simulator?
[1] Zhou et al, 2023. CodeBERTScore: Evaluating Code Generation with Pretrained Models of Code. https://arxiv.org/abs/2208.03133

---

> ### Author Response · Authors · 2023-11-16
>
> We thank the reviewer for a thorough review of our paper, enumerating its strongest points, asking us clarifying questions and providing new ways to test and evaluate our approach.
>
> ## **Questions:**
>
> **Q1: Can it scale to beyond egocentric videos in EPIC kitchen to third-person tutorial videos? How would the prompt change, especially in the phase 2 where the prompts and state-action predicates seems to be centered on the human in the videos?**
>
>
> While we did not evaluate our system on third-person videos in this paper, `Video2Demo` can easily be applied to generate state and action predicates for third-person tutorial videos.
>
> `Video2Demo`'s VLM and LLM prompts are mostly domain-independent. To gain better performance in new environments, one only needs to modify the following:
>
> * Domain information in the LLM's prompt, which could include
>     + Overall setting or theme of the vision language demonstrations (e.g. house chores)
>     + Examples of the state predicates
>     + Examples of the action predicates
>     + Domain-specific rules (e.g. we forbid the LLM from directly referring to the human in its state and action predicates).
> * Domain information in the VLM
>     + Whether the images are first-person view or third-person view
>
> **Q2: How does ChrF compare to other code generation metrics?**
>
> We thank the reviewer for suggesting CodeBERTscore as another code evaluation metric. We will integrate this in the final version of the paper and strengthen our evaluation of the generated code.
>
> **Does the generated code with high ChrF score correlate with human preference?**
>
> Our qualitative testing showed a correlation between generalized code with loops and relevant conditions, and a high ChrF score
>
> **How much of the generated code follow required syntax and physical feasibility to execute successfully on a simulator?**
>
> As we provide in the paper, the results from states and actions generated by baselines and `Video2Demo` imply that the demonstrations provided by our perception system will be noisy. However, we qualitatively show that the code generated is close to expert-generated code. In addition, we note that:
> 1. Demo2Code is robust to these noisy states and actions, and manages to capture the user’s preference
> 2. The code generated is valid Python code without any undefined functions
> As also mentioned in our global response, in future work, we will test the generated code for validity and feasibility in simulators like AI2-Thor [2] and VirtualHome [3].
>
> ## **Weaknesses:**
>
> ### **On more Evaluations**
> In our global response, we expand more on closing the loop with `Video2Demo` and evaluate using simulators [1-3]. We thank the reviewer for suggesting the ALFRED [1] benchmark to test our method. Since they also use high-level actions and egocentric vision, it provides the right testing ground. Our aim with this paper is to motivate the need for perception and generation of state-action sequences from task videos, and in future work, we hope to evaluate `Video2Demo` on the simulator.
>
> ### **On the reliance on prompt engineering**
>
> Most section of `Video2Demo`'s prompt is domain-independent, so it can be applied to other activity videos. For domains significantly different EPIC-Kitchens, the only parts of the prompts that need to be changed are about the domain. Please refer to our answer to your 1st question for details.
>
>
> ### References -
> [1] Mohit Shridhar, et al. ALFRED: A Benchmark for Interpreting Grounded Instructions for Everyday Tasks. arXiv:1912.01734, 2020
>
> [2] Eric Kolve et al. AI2-THOR: An Interactive 3D Environment for Visual AI. arXiv:1712.05474, 202
>
> [3] Xavier Puig et al. VirtualHome: Simulating Household Activities via Programs. arXiv:1806.07011, 2018
>
> [4] Zhou et al, 2023. CodeBERTScore: Evaluating Code Generation with Pretrained Models of Code. https://arxiv.org/abs/2208.03133

---

> > ### Author Response · Authors · 2023-11-21
> > **Thank you for your review**
> >
> > Hello!
> >
> > Thank you again for your feedback and questions, and suggestions to strengthen our paper!
> >
> > As per your suggestion, we have additionally evaluated code generated downstream using `Video2Demo`'s state-action predicates using CodeBERTScore and included it in the global response.
> >
> > As the discussion period ends soon, please let us know if we can provide any additional clarifications or answers that would help you in your evaluation.

---

> > > ### Comment · Reviewer_51er · 2023-11-22
> > > **Thanking authors for clarifications!**
> > >
> > > I would thank the authors for evaluating the baselines with other metrics and clarifications for all the reviewers' concerns. While the paper explores an interesting question of making use of LLM and VLM for video understanding, the proposed solution needs improvement in terms of scalability across different domains, and evaluation on sim/real robot setups. Even for offline video processing, the proposed solution seems to create one state/action predicate based on action segments labelled in EPIC-kitchen (response to reviewer 8paA) and it is unclear how an image will be selected as a state at test time.
> > > So, I would maintain my current score.

---

### Official Review · Reviewer_qcN7 · 2023-11-01

**Soundness:** 3 good
**Presentation:** 4 excellent
**Contribution:** 2 fair
**Rating:** 3
**Confidence:** 4

**Summary:**

The paper aims to propose a model that can extract temporal-consistent (text) state and (text) action pairs from videos. In order to do so, the paper proposes the let VLM and LLM talk with each other to extract descriptions. Since LLM can see past communications and can be prompted to take VLM output critically, the extracted description can consistently track objects. To evaluate the method, the authors provides a human annotated state-action predicates for EPIC-Kitchens. Finally, the authors show that LLM can use such extraction as demonstrations to prompt LLM to synthesize robot code.

**Strengths:**

- The paper is well written and is easy to understand.

- The paper not only shows the perception power but also demonstrates downstream applications like code synthesis and planning.

- The paper contributes a small human-annotated validation set for EPIC-Kitchens, a nice contribution to the community who hopes to do similar work.

- All design choices are logical and sounds.

**Weaknesses:**

- My main criticism for this paper is its contribution's significance. Using LLM and VLM with text as history seems like an obvious design choice. The techniques the authors introduced over the Socratic Model, namely the way to structure and prompt the LLM / VLM interaction doesn't seem to constitute enough contribution to be an ICLR paper.

- The evaluation itself relies on GPT, which is a bit weak despite the human annotation the authors provide. If the authors had proposed a structured output format like those used in VQA and have more annotations the evaluation would be much stronger.

- There are also a few misleading claims. Throughout the paper, the authors talks about constructing "state-action" pairs, while in reality what they extract are some loose-form text predicates as well as loose-form text actions. Such abusive use of terms misleads the readers when they read the abstract.

Overall, I think the current status of the paper lacks the significance an ICLR paper would need.

**Questions:**

1. How big is the "Human-annotated state and action predicates" in claimed contribution 2? I think this is a nice contribution but from what I read in the paper, this doesn't seem to be very big. Could you clarify?

2. Could you clearly define "structure and temporal consistency" in the abstract?

---

> ### Author Response · Authors · 2023-11-16
>
> We thank the reviewer for their comprehensive review commending the strengths of the paper as well as constructive criticism about the contributions.
>
> ### **Questions**
>
> **Q1: How big is the "Human-annotated state and action predicates" in claimed contribution 2? I think this is a nice contribution but from what I read in the paper, this doesn't seem to be very big. Could you clarify?**
>
> We annotate 20 videos from the EPIC-Kitchens validation dataset, where each video is 5 minutes in length on average
> * an average of 3 predicates per frame
> * 1 action per frame, which is already provided by the dataset.
> * 1027 image frames in total
>
> We also partially annotate 5 videos to only focus on the dish-washing segment of the videos.
> All annotations are double-checked by a second person to ensure quality.
>
> We estimated that the time required for manual labeling per video is 2 hours, resulting in a total estimated annotation time of 50 hours. Thus, we built a user-friendly web-based annotation tool that allows users to cross-reference frames and autocomplete predicates based on past annotation and reduces annotation time.
>
> **Q2: Could you clearly define "structure and temporal consistency" in the abstract?**
>
> The term structure and temporal consistency refers to the importance of understanding past actions and the relevance of objects in past timesteps (i.e. state of the environment) to predict state-actions for the current timestep. Referring to past state-actions to infer the current state and action is evident during video annotations as human annotators as well. For example, if a cook is seen holding a pan in their hand at some time t:
> * If the pan was brought over to the stove-top from elsewhere, it is likely to be put down in the next few frames
> * If the pan was not in hand earlier, it was picked up right before time t.
> * The pan, as part of the environment, may stay relevant in future actions in the demonstration.
>
> If we use a VLM to generate captions without context, it may fail to recognize what is going on in the scene. However, responses from VLMs along with GPT-4’s ability to reason what states precede actions and what actions follow other actions enhances the consistency in actions and the state of the observed environment.
>
> ## **Weaknesses**
>
> ### **On the contribution in the paper**
>
> The contribution of the paper is not just in the method of LLM/VLM interaction, but also in solving the problem of extracting symbolic states and actions from long vision language demonstration. The baseline result shows that
> * The captions that VLM generates are not sufficient in extracting salient states about objects in the frame. (see SM-Caption)
> * Despite including history, a predefined, fixed interaction between the LLM and the VLM is also not sufficient in accurately extracting salient information because the LLM has no way to query VLM about inconsistencies in its answer. (see SM-Fixed-Q)
>
> Naive application of LLM and VLM cannot effectively solve this problem. `Video2Demo`, despite its simplicity, is able to significantly outperform the baselines by extracting symbolic states and actions with higher quality.
>
> ### **Clarification on Evaluation**
>
> We use GPT-3.5 to evaluate our method’s generated state and action predicates as they come from:
> 1. open-vocabulary based predicate generation and
> 2. expert annotations from different humans
> meaning that it's possible two predicates match semantically without matching exactly in text string. Since it is taxing to enumerate all possible states in a complex environment like EPIC-Kitchens, we allow GPT-4 to come up with its own predicates. We argue that in this setting a soft match in the semantic space is fair to evaluate the model. In addition to this, a natural-language planner should be able to generate similar plans for semantically similar predicates.
>
> In order to make the evaluation more robust, we ask GPT-3.5 to give reasoning before giving an answer (of whether two predicates are “equal” in semantic space)
>
> Text evaluations are often done using other AI models, especially more recently with reliable models like GPT3.5 or GPT4. [1,2]
>
> ### **Clarification on Outputs**
> In the global response under “Representation of States and Actions”, we clarify the choice of how we represent states and actions. While they may be text-based, they are structured in predicate format, and a majority of the most commonly used state predicates and actions are provided to GPT-4 as part of its prompt. Examples of predicates:
>
> * state: `is_inside(<A>, <B>)`, `human_is_holding(<A>)`, `is_on_top_of(<A>, <B>)`, etc.
> * action: `pick_up(<A>)`, `open(<A>)`, `turn_off(<A>)`, etc.
>
> Since it is taxing to design a comprehensive pddl-like specification for every possible predicate in a complex setting like EPIC-Kitchens, we allow the model to come up with its own predicates similar to the examples. This can be restricted in the prompt if `Video2Demo` is to be used in a simpler enumerable setting.

---

> > ### Author Response · Authors · 2023-11-16
> >
> > (continued)
> >
> > ### References -
> >
> > [1] Chiang, Cheng-Han, and Hung-yi Lee. "Can Large Language Models Be an Alternative to Human Evaluations?." arXiv preprint arXiv:2305.01937 (2023).
> >
> > [2] Mao, Rui, et al. "GPTEval: A survey on assessments of ChatGPT and GPT-4." arXiv preprint arXiv:2308.12488 (2023).

---

> > > ### Author Response · Authors · 2023-11-21
> > > **Thank you for your review**
> > >
> > > Hello!
> > >
> > > Thank you again for your feedback, questions, and suggestions to strengthen our paper!
> > >
> > > As the discussion period ends soon, please let us know if we can provide any additional clarifications or answers that would help you in your evaluation.

---

> ### Comment · Reviewer_qcN7 · 2023-11-23
>
> Thank you for the clarifications.
>
> I just went through the response written by peer reviewers, as well as the author's response. Unfortunately, I don't think the author's rebuttal would address my concerns, without heavy modifications to the project. I reiterate the need for stronger technical contribution for this paper to be accepted at any top ML conference.

---

### Official Review · Reviewer_8paA · 2023-11-02

**Soundness:** 2 fair
**Presentation:** 2 fair
**Contribution:** 1 poor
**Rating:** 3
**Confidence:** 3

**Summary:**

The authors propose using a video language model and a large language model in tandem to label videos from the Epic-Kitchens dataset with descriptions of the various subtasks demonstrated in the video in pseudo-code. They evaluate the ability of their method on this task using a handful of videos with hand-labeled pseudo-code descriptions.

**Strengths:**

**Well-written**: The paper is clear with good presentation, sound descriptions of the idea and clear experiments. The authors clearly state their setups.

**Weaknesses:**

Unfortunately, I am not fully convinced of the motivation behind this work. I enlist the weaknesses of this work below:
1. **Not enough evaluation**: it's very difficult to get a signal for the method's abilities given that evaluations are performed on contrived code for 7 videos.
2. **Choice of action space perhaps makes this task too easy**: Generating code is generally useful but in this setup, it's difficult to apply to real-life scenarios due to the level of abstraction. For instance, the reference code uses functions like “check_if_dirty(object)”, a level of abstraction for which we do not have good robot behaviors. In a sense, the method performs a kind of task-level planning. But the level of abstraction of this planning makes it impossible to test in a control setup.
3. **Unclear motivation**: The authors claim that this work tries to ground videos into state actions and states for robot demos. Unfortunately, this is simply not true. They describe the state in videos using text and ground actions into pseudo-code. This far from the promise of a state-action demonstration.
4. **No robotic evaluations**: The paper does not run any experiments on robots - neither in simulation nor on real robots. Therefore, I believe calling this a method to generate state-action demonstrations is an overclaim.
5. **Extremely expensive to deploy on a robot**: The method requires making several calls per time step of execution to GPT-4 making this very very expensive.

**Questions:**

1. What was the cost of running these experiments?
2. Do you have one state for every image?

---

> ### Author Response · Authors · 2023-11-16
>
> We thank the reviewer for their constructive criticism, and we acknowledge their concerns about our paper’s motivation, evaluation, and practical deployment.
>
> ## **Questions:**
>
> **Q1: What was the cost of running these experiments?**
>
> The costs for these experiments are mainly the cost of OpenAI’s GPT-4 API usage.
>
> * The total cost was approximately $2200.
> * Each demonstration costs about $15.
>
> We ran approximately 6 experiments on each of the 26 videos, encompassing several trials with the baselines, `Video2Demo`, ablations, and qualitative experiments for code generation.
>
> Lower costs for GPT-4 were recently announced by OpenAI [6], and we can accordingly expect the cost of this method to be reduced in the future.
>
>
> **Q2:Do you have one state for every image?**
>
> We do not generate states for every frame in the video because two consecutive frames are highly similar. Instead, `Video2Demo` generates a set of state predicates and one action for each action segment specified in the EPIC-Kitchens data.
>
>
> ## **Weaknesses**
>
> ### **On insufficient evaluation**
>
> In quantitative evaluation, we focus on the quality of the generated state and action predicates. In Table 1 and Figure 3 in our paper, we evaluated the approaches on 19 videos that have human state-action annotation. We show how our generated predicates are closest to human annotations compared to other baselines across all kitchen activity categories.
>
> Our evaluation for code generation on 7 dishwashing videos is a qualitative example of a downstream application of `Video2Demo`. We show that [2], which takes text-based state-action trajectories as input, can generate expert-like code when it uses the state-actions generated by `Video2Demo`.
>
> ### **Clarification on the choice of action space**
>
> In global response, we clarify why we picked a high-level action space. We would like to add that despite the high-level resolution of this task, predicting these high-level actions is still not easy, as is evident from the accuracy and recall scores from the baseline VLM-based methods and even `Video2Demo`.
>
> While we acknowledge that code generated by task planners may not be feasible or affordable by the embodiment in that environment, we note that it is not the focus of our work. We aim to be able to generate states and actions that can increase the success rate for language-based planners compared to only providing language instructions. Qualitatively, we see that our demonstrations allow a code-generation pipeline based on [1-2] to generate expert-like code with loops and conditions that are non-trivial to infer from simple language instructions.
>
>
> ### **Clarification on Motivation**
>
> In the global response, we clarify our motivation behind this work and why we represent our state-action sequences in text-predicate form.
>
> Our method aims to ground the visual information available in hours of offline demonstration data such as EPIC-Kitchens by converting it into text-based states and actions. We hope that language-based task planners [2-5] can learn the high-level task plan from symbolic state-action trajectory—bridging the gap between language instruction and planning, as seen in [1-2].
> ### **On Robotic Evaluation**
>
> In the global response, we elaborate on how the paper focuses on the problem of extracting rich information from offline vision-language demonstrations. In future work, we plan to evaluate downstream applications of `Video2Demo`’s generated state-action in simulations (e.g. ALFRED, AI2-THOR [7-8]) and on real robots.
> On Cost and Speed of Deployment
> We acknowledge the importance of fast perception models during the deployment and online rollout of learned policies. However, **`Video2Demo`'s applications are for offline settings, so it does not need to be deployed on a robot.**
>
> Concretely, `Video2Demo` provides a framework for learning high-level task plans from few-shot demonstrations. We hope that with future work, we can:
> 1. Collect one or more demonstrations of a human performing a task.
> 2. Run `Video2Demo` to extract the high-level states and actions from these demonstrations
> 3. Rely on task-planning and motion-planning methods to leverage these generated states and actions to replicate the tasks, e.g., generate code to perform this task, conditioned on the robot embodiment, environment affordances etc.
>
> Thus, `Video2Demo` can run offline asynchronously before the policy is learned then run on the robot.

---

> > ### Author Response · Authors · 2023-11-16
> >
> > (continued)
> >
> > ### References -
> >
> > [1]Jacky Liang, et al. Code as policies: Language model programs for embodied control
> >
> > [2] Wang, Huaxiaoyue et al. Demo2Code: From Summarizing Demonstrations to Synthesizing Code via Extended Chain-of-Thought. arXiv:2305.16744, 2023
> >
> > [3] Wenlong Huang et al. VoxPoser: Composable 3D Value Maps for Robotic Manipulation with Language Models. arXiv:2307.05973, 2023
> >
> > [4] Michael Ahn et al. Do As I Can, Not As I Say: Grounding Language in Robotic Affordances. arXiv:2204.01691, 2022
> >
> > [5] Danny Driess et al. PaLM-E: An Embodied Multimodal Language Model. arXiv:2303.03378, 2023
> >
> > [6] OpenAI: New models and developer products announced at DevDay. https://openai.com/blog/new-models-and-developer-products-announced-at-devday, 2023
> >
> > [7] Mohit Shridhar, et al. ALFRED: A Benchmark for Interpreting Grounded Instructions for Everyday Tasks. arXiv:1912.01734, 2020
> >
> > [8] Eric Kolve et al. AI2-THOR: An Interactive 3D Environment for Visual AI. arXiv:1712.05474, 2022

---

> > > ### Author Response · Authors · 2023-11-21
> > > **Thank you for your review**
> > >
> > > Hello!
> > >
> > > Thank you again for your feedback, questions, and suggestions to strengthen our paper!
> > >
> > > As the discussion period ends soon, please let us know if we can provide any additional clarifications or answers that would help you in your evaluation.

---

### Author Response · Authors · 2023-11-16

We thank the reviewers for their thorough and thoughtful feedback on our paper.
We are pleased to read that reviewers found the paper easy to read and the setup easy to follow (8paA, qcN7, 51er, RP75), acknowledge our motivation behind building a perception system like this (qcN7, RP75), the novelty in tackling a state-action generation task (51er), and finally the potential and challenges in building pipelines with AI systems interacting with each other (51er).

We would like to preface our response with a summary of our motivation and our contributions.

### **Overall Goal of the Paper**

Our goal is to automatically extract salient symbolic states and actions about a task from an offline vision-language demonstration.

The extracted symbolic states and actions can be used in various downstream applications:
1. The states and actions provide a high-level representation of the demonstration, which can be summarized to learn the high-level task that the user is demonstrating [1-2].
2. We can auto-generate states and actions from demonstration videos and build datasets for training action prediction modules that are deployed on real robots.
3. Recent work, that focuses on language-based task planning, often lacks personalization and does not capture human preferences [3-5]. State-action demonstrations of the task can help capture these preferences and make task planning easier.

These applications can be tested in simulators like ALFRED, VirtualHome, etc. [6-8], which work in a similar high-level task planning domain.

In our paper, **we focus on quantitatively evaluating** the quality of states and actions generated by `Video2Demo`. We systematically compare `Video2Demo`'s output against human state-action annotations of EPIC-Kitchen's egocentric real-world videos of users doing tasks in their own kitchens. While we also evaluate downstream applications like code generation, we note that the primary evaluation is predicting correct state actions. As our results show, this is a challenging benchmark where current VLMs struggle. We believe our approach would serve as a simple but valuable baseline.


### **Main Contributions**

1. We are the first method to extract symbolic states and actions from long-horizon vision language demonstrations for downstream robot task planning. We believe that our approach can easily plug into existing LLM-based task planners to learn new tasks from vision language demonstrations.
2. We provide human annotations in text-based state predicates for 26 videos from the EPIC-Kitchens validation dataset. There are 1027 annotated image frames with roughly 3 state predicates per frame. We use the existing action annotations from the dataset.
3. Our method outperforms baselines that naively combine generative VLMs and LLMs in both state and action prediction.

We also address some common questions and feedback.

### **States and Actions Representations**

The main focus of this paper is to generate states and actions that can help a task planner understand high-level task plans shown in the demonstration. This goal influenced our decision to represent states and actions as open-set symbolic predicates (e.g. `in_hand('can')`, `open('shelf')`). Concretely,
* Given a small set of example state and action predicates, Video2Code's LLM can easily generalize and define new predicates that are suitable for an unseen vision language demonstration.
* A task planner that uses LLMs can easily reason about text-based predicates and learn the corresponding high-level task plan. Its LLMs can also robustly handle predicates that are semantically similar but syntactically different (e.g. `in_hand('can')` v.s. `is_holding('can')`).

### **Evaluation on robots**

We acknowledge that `Video2Demo`'s effect on downstream applications is best shown through real-robot or simulator experiments.

However, this paper focuses on a less-explored problem of extracting text-based states and actions from a rich vision language demonstration. After directly evaluating the correctness of `Video2Demo`'s generated states and actions against human annotation, we demonstrate how `Video2Demo`'s output can be used by downstream applications [2] to generate high-level task plans as code.

For future work, we aim to integrate this with a simulator [6-8], which has high-level task planning settings, and evaluate how the generated code can successfully imitate the provided demonstration.

---

> ### Author Response · Authors · 2023-11-16
>
> (continued)
>
> ### References -
>
> [1] Huaxiaoyue Wang, et al. Demo2Code: From Summarizing Demonstrations to Synthesizing Code via Extended Chain-of-Thought. arXiv:2305.16744, 2023
>
> [2] Wu, Jimmy, et al. "Tidybot: Personalized robot assistance with large language models." arXiv preprint arXiv:2305.05658, 2023
>
> [3] Wenlong Huang, et al. Voxposer: Composable 3d value maps for robotic manipulation with language models. arXiv:2307.05973, 2023
>
> [4] Tianbao Xie, et al. Text2Reward: Automated dense reward function generation for reinforcement learning. arXiv:2309.11489, 2023
>
> [5] Yu, Wenhao, et al. "Language to Rewards for Robotic Skill Synthesis." arXiv:2306.08647, 2023
>
> [6] Eric Kolve et al. AI2-THOR: An Interactive 3D Environment for Visual AI. arXiv:1712.05474, 2022
>
> [7] Xavier Puig et al. VirtualHome: Simulating Household Activities via Programs. arXiv:1806.07011, 2018
>
> [8] Mohit Shridhar, et al. ALFRED: A Benchmark for Interpreting Grounded Instructions for Everyday Tasks. arXiv:1912.01734, 2020

---

> ### Author Response · Authors · 2023-11-20
> **Additional Evaluations**
>
> We once again thank reviewer 51er for suggesting we use CodeBERTScore[9] to additionally evaluating the quality of the code generated by `Video2Demo` using [1].
>
> The following table summarizes the qualitative evaluation of code generated using our baselines, `Video2Demo` and the human-annotated state-action demonstrations, as described in Table 2 in our paper, replacing ChrF with codeBERTScore as the metric.
>
> |      Model                    | P30-07 | P22-05 | P04-101 | P07-10 | P22-07 | P30-08 | P7-04 | Average | Action Acc | State Rec |
> | ----------------------------- | ------ | ------ | ------- | ------ | ------ | ------ | ----- | -------- | ---------- | --------- |
> | **SM-caption**                | 0.839  | 0.787  | 0.897   | 0.811  | 0.809  | 0.879  | 0.902 | 0.846    | 13.10      | 20.48     |
> | **SM-fixed-q**                | 0.854  | 0.762  | **0.943**| 0.853  | **0.849**| **0.917**| 0.897| 0.868    | 18.31      | 35.47     |
> | **Video2Demo**                | **0.956**| **0.866**| 0.852   | **0.910**| 0.822  | 0.847  | **0.912**| **0.881**| **20.25** | **59.27**  |
> | **annotated-demo***           | 0.851  | 0.975  | 0.911   | 0.961  | 0.965  | 0.942  | 0.908 | 0.930    | -          | -         |
>
> We see that codeBERT generally agrees with ChrF scores, and favors code written by `Video2Demo` over baselines.
>
> added reference -
> [9] Zhou, Shuyan, et al. "Codebertscore: Evaluating code generation with pretrained models of code." arXiv preprint arXiv:2302.05527 (2023).

---

> > ### Author Response · Authors · 2023-11-21
> > **Comparisons with GPT4-V**
> >
> > We thank reviewer RP75 for suggesting comparison against GPT4-V as a VLM to generate state-action demonstrations. We conduct additional experiments using OpenAI's newly released `gpt-4-vision-preview` model[10], however due to current request limitations, we qualitatively compare our method to GPT4-V on two videos from the EPIC-Kitchens dataset. We also ablate by holding-out the list of objects in the scene that we provided to `Video2Demo`.
> >
> > ### P18_02
> > | Model               | Action Accuracy | State Recall |
> > |---------------------|-----------------|--------------|
> > | SM-Caption         | 8.0%               | 9.7%         |
> > | SM-Fixed-Q         | 20.0%              | 33.3%          |
> > | Video2Demo          | **24.0%**      | **50%**         |
> > | GPT4-V (no objects)  | **24.0%**           | 29.0%          |
> > | GPT4-V (with objects)  | 16.0%              | 33.3%          |
> >
> > ### P03_22
> > | Model               | Action Accuracy | State Recall |
> > |---------------------|-----------------|--------------|
> > | SM-Caption          | 4.2%             | 8.0%           |
> > | SM-Fixed-Q         | 33.3%              | 36.0%          |
> > | Video2Demo          | 33.3%              | 56.0%          |
> > | GPT4-V (no objects)  | 13.0%               |  17.0%            |
> > | GPT4-V (with objects)  | **50.0%**             | **64.0%**         |
> >
> > We see that despite slight improvements in one demonstration, GPT4-V is not immune to hallucinations and performs inconsistently across both tests.
> >
> > Due to API, time and cost constraints, we could not use GPT4-V as a VLM in our pipeline, however, we argue that a stronger perception module, even with hallucinations, will improve our pipeline. Our pipeline better grounds the perception module with open-ended reasoning, feedback and self-critical behavior that can correct mistakes in the generated response. This is akin to several recent works in self-correcting AI[11,12].
> >
> > added references -
> >
> > [10] OpenAI: Vision. https://platform.openai.com/docs/guides/vision, 2023
> >
> > [11] Madaan, Aman, et al. "Self-refine: Iterative refinement with self-feedback." arXiv preprint arXiv:2303.17651 (2023).
> >
> > [12] Bai, Yuntao, et al. "Training a helpful and harmless assistant with reinforcement learning from human feedback." arXiv preprint arXiv:2204.05862 (2022).

---

### Meta-Review · Area_Chair_eT2N · 2023-12-04

**Metareview:**

This work combined multiple large foundation models by using GPT-4 to query a generative VLM to construct state-action sequences which are passed into a language model to generate robot task code. The strengths of this work include a novel framing of decoding video content as an iterative dialog between VLMs and LLMs. The paper is well-structured and easy to understand, making it accessible to a broad audience. It effectively communicates key research questions and failure cases, bringing attention to spatial grounding and hallucinations in LLMs and VLMs. However, the lack of simulated or real robot experiments makes it challenging to assess the practicality of the proposed approach and identify possible failure scenarios. The paper could benefit from discussions on how the generated task plans would perform in simulated environments. The reliance on prompt engineering, while effective for EPIC kitchen videos, raises questions about the applicability of the presented prompts to other activity videos. The effort involved in prompt engineering is not fully addressed. Additionally, the paper lacks a comparison with other LLMs and VLMs and does not discuss how the framework generalizes to alternative models. Somewhat contradictory to the last point, but the current price tag for these experiments is also a little shocking - $15 per demonstration costs more than a human labeler, although it is noted that this could go down in the future.

**Justification For Why Not Higher Score:**

I am concerned by the limited novelty and impracticality of this work, as well as the lack of proper evaluation on control environments. Compared to the amount of effort and compute that went into extracting demonstrations, it's a little concerning that evaluation in simulation environments was not included.

**Justification For Why Not Lower Score:**

N/A

---

### Decision · Program_Chairs · 2024-01-16

Reject